# ACCELERATING 3D MOLECULE GENERATION VIA JOINTLY GEOMETRIC OPTIMAL TRANSPORT

**Haokai Hong, Wanyu Lin,**[*] **Kay Chen Tan**
Department of Data Science and Artificial Intelligence
Department of Computing
The Hong Kong Polytechnic University, Hong Kong SAR, China.
`haokai.hong@connect.polyu.hk`, `{wan-yu.lin,kctan}@polyu.edu.hk`

## ABSTRACT

This paper proposes a new 3D molecule generation framework, called GOAT, for fast and effective 3D molecule generation based on the flow-matching optimal transport objective. Specifically, we formulate a geometric transport formula for measuring the cost of mapping multi-modal features (e.g., continuous atom coordinates and categorical atom types) between a base distribution and a target data distribution. Our formula is solved within a joint, equivariant, and smooth representation space. This is achieved by transforming the multi-modal features into a continuous latent space with equivariant networks. In addition, we find that identifying optimal distributional coupling is necessary for fast and effective transport between any two distributions. We further propose a mechanism for estimating and purifying optimal coupling to train the flow model with optimal transport. By doing so, GOAT can turn arbitrary distribution couplings into new deterministic couplings, leading to an estimated optimal transport plan for fast 3D molecule generation. The purification filters out the subpar molecules to ensure the ultimate generation quality. We theoretically and empirically prove that the proposed optimal coupling estimation and purification yield transport plan with non-increasing cost. Finally, extensive experiments show that GOAT enjoys the efficiency of solving geometric optimal transport, leading to a double speedup compared to the sub-optimal method while achieving the best generation quality regarding validity, uniqueness, and novelty. The code is available at github.

## 1 INTRODUCTION

The problem of 3D molecule generation is essential in various scientific fields, such as material science, biology, and chemistry (Hoogeboom et al., 2022; Watson et al., 2023; Xie et al., 2022). Typically, 3D molecules can be represented as atomic geometric graphs (Hoogeboom et al., 2022; Xu et al., 2023; Song et al., 2023b), where each atom/node is embedded in the Cartesian coordinates and encompasses rich features, such as atom types and charges. There has been fruitful research progress on geometric generative modeling for facilitating the process of 3D molecule generation. Specifically, geometric generative models are proposed to estimate the distribution of complex geometries and are used for generating feature-rich geometries. The success of these generative modeling mainly comes from the advancements in using the notion of probability measurement transport for generating samples. Generative modeling aims to generate samples via mapping a simple prior distribution, e.g., Gaussian, to a desired target probability distribution. This mapping process can be termed as a *distribution transport/generative problem* (Peluchetti, 2023).

Recent representative models for sampling 3D molecules in silicon include diffusion-based models (Hoogeboom et al., 2022; Wu et al., 2022; Xu et al., 2023) and flow matching-based models (Song et al., 2023a; Klein et al., 2023). Diffusion-based models have shown superior results on molecule generation tasks (Xu et al., 2023; Hoogeboom et al., 2022; Jung et al., 2024). They simulate a stochastic differential equation (SDE) to transport a base distribution (*e.g.*, Gaussian) to the data distribution. However, a major drawback of diffusion-based models is their slow inference

---

[*]Corresponding author.

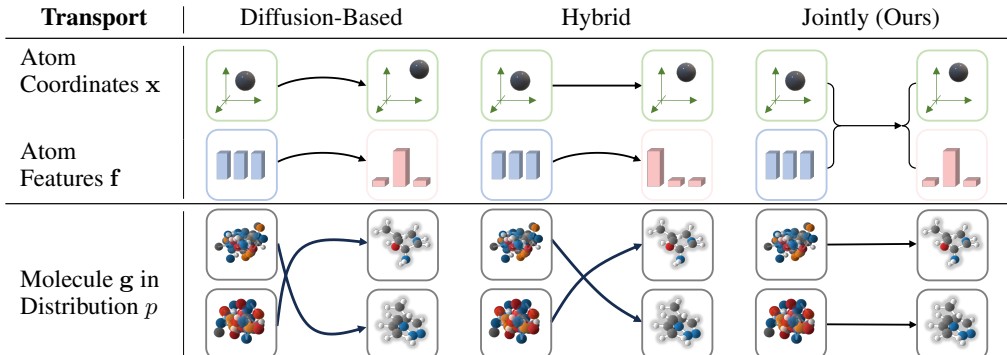

Figure 1: **The Illustration of Probability Paths based on Different Molecule Generative Models.**
1. The diffusion path (Hoogeboom et al., 2022; Xu et al., 2023), which typically deviates from a
straight line map, necessitates a large number of sampling steps. 2. The hybrid transport (Song
et al., 2023a) ensures straight transport for atomic coordinates, but it does not guarantee the same for
atom features. Furthermore, this cost does not consider the optimal distribution couplings, leading to
suboptimal transport between distributions. 3. GOAT simultaneously considers the optimal transport
for atom coordinates and features, providing a joint and straight path for fast sampling.

speed with the learned stochastic transport trajectory (Hoogeboom et al., 2022; Wu et al., 2022;
Xu et al., 2023); they typically need approximately $1,000$ sampling steps to produce valid samples.
This could make large-scale inference prohibitively expensive. Accordingly, flow matching — built
upon continuous normalizing flows — has emerged as a new paradigm that could potentially provide
effective density estimation and fast inference (Lipman et al., 2022; Tong et al., 2023; Dao et al.,
2023; Liu et al., 2022; Klein et al., 2023; Song et al., 2023a).

*This paper aims to deal with the problem of fast and effective 3D molecule generation based on flow
matching optimal transport principle.* In particular, our objective is to obtain a distribution transport
trajectory with optimal transport cost and generation quality regarding molecule validity, unique-
ness, and novelty. A few recent works have been proposed to improve the sampling efficiency of
the geometrical domains via flow matching principle. (Klein et al., 2023) was proposed for efficient,
simulation-free training of equivariant continuous normalizing flows, which can produce samples
from the equilibrium Boltzmann distribution of a molecule in Cartesian coordinates. However, it
can only work for many-body molecular systems and does not consider atomic features.

In the context of molecule generation, properly characterizing the transport cost to optimize over is
indispensable and challenging. There are two main challenges. *Firstly*, the multi-modal property
of molecular space, typically consisting of continuous atom coordinates and categorical atom types,
makes the transport cost measurement hard to optimize. *Secondly*, the optimal transport problem
essentially is to search optimal distribution couplings for mapping. This process typically requires
similarity computation of the two distributions. However, the various sizes of geometric graphs
to transport introduce difficulties in evaluating the distribution similarity. The closest to ours is
EquiFM (Song et al., 2023a), which attempts to address the multi-modality issue by using different
probability paths to transport each modality separately. The proposed equivariant optimal transport
(OT) for transporting atom coordinates indeed forms a straight-line trajectory for training, while the
variance-preserving principle could not ensure a straight-line trajectory for atom features. Therefore,
the fused flow paths might deviate strongly from the OT paths and could not ensure optimal coupling
between two probability measurements, leading to high computational costs and numerical errors.

In this work, we propose a new framework for fast 3D molecule generation based on a novel and
principled optimal transport flow-matching objective, dubbed as **G**eometric **O**ptim**A**l **T**ransport
(GOAT). In particular, we formulate a geometric transport cost measurement for optimally transport-
ing continuous atom coordinates and categorical features, which is inherently a Bilevel optimization
problem. To deal with the first challenge induced by transporting multiple modalities, GOAT lever-
ages a latent variable model equipped with equivariant networks to map the multi-modal features
into a joint, equivalent, and smooth representation space. This equivariant latent variable model has
been proven to be flexible and expressive for modeling complex 3D molecules (Satorras et al., 2021;
Xu et al., 2023). A latent flow matching then operates over the latent space, which can provide
distributional coupling estimation.

To tackle the second challenge — obtaining the optimal distribution couplings, we propose to refrain from directly working with distribution similarity computation. Specifically, we propose to rectify the flow with the estimated ones based on the latent flow matching. Because the estimated distributional couplings are identified based on the synthesized samples, they might deviate from the real-world samples in terms of quality. We provide a purification process for high-quality samples regarding validity, uniqueness, and novelty. With this process, we can turn arbitrary couplings into deterministic and causal ones, leading to the optimal transport path for fast and effective generation.

We theoretically demonstrate that optimal coupling estimation and purification result in non-increasing geometric transport costs. Moreover, we empirically highlight the superiority of GOTA by conducting experiments on widely used benchmarks. The proposed method reduced the transport cost by nearly 89.65%, halving the sampling times compared to EquiFM. In terms of generation quality, our method achieves up to a 17.1% improvement over existing algorithms.

## 2 PROBLEM SETUP

**Notations.** A three-dimensional (3D) molecule with $N$ atoms can be represented as a geometric graph denoted as $\mathbf{g} = \langle \mathbf{x}, \mathbf{h} \rangle$, where $\mathbf{x} = (\mathbf{x}_1, \dots, \mathbf{x}_N) \in \mathbb{R}^{N \times 3}$ represents the atom coordinates and $\mathbf{h} = (\mathbf{h}_1, \dots, \mathbf{h}_N) \in \mathbb{R}^{N \times d}$ is the atom features containing atomic types, charges, etc. $d$ is the dimensionality of the atom features. A zero center-of-mass (Zero CoM) space is defined as $\mathbb{X} = \{\mathbf{x} \in \mathbb{R}^{N \times 3} : \frac{1}{N} \sum_{i=1}^{N} \mathbf{x}^i = \mathbf{0}\}$, indicating that the mean of the $N$ atoms' coordinates should be 0. In what follows, we will introduce some necessary concepts, including flow matching and optimal transport, to facilitate the definition of our problem.

**General Flow Matching[1].** Given noise $\mathbf{x}_0 \in R^N \sim p_0$ and data $\mathbf{x}_1 \in R^N \sim p_1$, the general flow-based model considers a mapping $f : \mathbb{R}^N \to \mathbb{R}^N$ as a smooth time-varying vector field $u : [0, 1] \times \mathbb{R}^N \to \mathbb{R}^N$, which defines an ordinary differential equation (ODE): $d\mathbf{x} = u_t(\mathbf{x})dt$. Continuous normalizing flows (CNFs) were introduced with black-box ODE solvers to train approximate $u_t$. However, CNFs are hard to train as they need numerous evaluations of the vector field.

**Flow matching** (Lipman et al., 2022), a simulation-free approach for training CNFs, is proposed to regress the neural network $v_\theta(\mathbf{x}, t)$ to some target vector field $u_t(\mathbf{x})$: $\mathcal{L}_{\text{FM}}(\theta) := \mathbb{E}_{t \sim \mathcal{U}(0,1), \mathbf{x} \sim p_t(\mathbf{x})} \|v_\theta(\mathbf{x}, t) - u_t(\mathbf{x})\|^2$, where $p_t(\mathbf{x})$ is the corresponding probability path which conditioned on $\mathbf{x}_1 \sim p_1$ and then defined as $p_t(\mathbf{x}|\mathbf{x}_1) = \int p_t(\mathbf{x}|\mathbf{x}_1)p_1(\mathbf{x}_1)d\mathbf{x}_1$. In implementation, common probability paths include variance exploding (VE) diffusion path (Song et al., 2020), variance preserving (VP) diffusion path (Ho et al., 2020), and straight transport path (Lipman et al., 2022; Liu et al., 2022).

**Optimal Transport (OT).** Transport plan between $p_0$ and $p_1$ is also called coupling and we denoted it as $\Gamma(p_0, p_1)$ (Ambrosio et al., 2021). OT addresses the problem of finding the optimal coupling that minimizes the transport cost, typically in the form of $\mathbb{E}[c(\mathbf{x}_1 - \mathbf{x}_0)]$, where $c : \mathbb{R}^N \to \mathbb{R}$ is a cost function, such as $c(\cdot) = \|\cdot\|^2$. Formally, a coupling $\Gamma(p_0, p_1)$ is called optimal only if it achieves the minimum value of $\mathbb{E}[c(\mathbf{x}_1 - \mathbf{x}_0)|(\mathbf{x}_1, \mathbf{x}_0) \in \Gamma(p_0, p_1)]$ among all couplings that share the same marginals. An ideal optimal transport trajectory is a map of optimal couplings.

**Geometric Optimal Transport.** In our task, we consider a pair of geometric probability distributions [2] over $\mathbb{R}^{N \times (3+d)}$ with densities $p(\mathbf{g}_0)$ and $p(\mathbf{g}_1)$ (or denoted as $p_0$ and $p_1$). Geometric generative modeling considers the task of fitting a mapping $f$ from $\mathbb{R}^{N \times (3+d)}$ to $\mathbb{R}^{N \times (3+d)}$ that transforms $\mathbf{g}_0$ to $\mathbf{g}_1$. More specifically, if $\mathbf{g}_0$ is distributed with density $p_0$ then $f(\mathbf{g}_0)$ is distributed with density $p_1$. Typically, $p_0$ is an easily sampled density, such as a Gaussian.

In our specific problem, beyond geometric distribution transport, we concentrate on fast 3D molecule generation based on the flow model with optimal transport (OT), which has been proven effective in accelerating non-geometric flow models (Liu et al., 2022; Tong et al., 2023). Therefore, our problem is defined as geometric optimal transport flow matching.

---

[1]We are aware of the drawbacks of reusing the notation $\mathbf{x}$, which represents a general data point here.

[2]Geometric probability distribution denotes the molecular data distribution and it is to reflect the geometric property of molecules as opposed to non-geometric data, such as text.

# 3 OUR METHOD: GOAT

## 3.1 FORMULATING GEOMETRIC OPTIMAL TRANSPORT PROBLEM

Our objective is to obtain an optimal transport trajectory for fast 3D molecule geometry generation based on optimal transport flow models. In this regard, we can reformulate our problem into searching for optimal coupling for geometric optimal transport. Specifically, it involves the transportation of molecules via optimal coupling, where each atom follows a straight and shortest path. In other words, each molecule is coupled with a noisy sample that incurs the minimum cost, and each atom within the target molecule is paired with its counterpart in the noise, leading to the minimum cost. To encapsulate the optimal scenario, we consider the problem of geometric optimal transport with two components: 1) *optimal molecule transport (OMT) with equivariant OT for atom coordinates and invariant OT for atom features*; 2) *optimal distribution transport (ODT) with optimal molecule coupling*.

**Geometric Transport Cost.** Transporting a molecule includes transforming atom coordinates and features. We can depict the molecule transport cost as below:

$$c_g(\mathbf{g}_0, \mathbf{g}_1) = \|\mathbf{x}_1 - \mathbf{x}_0\|_2 + \|\mathbf{h}_1 - \mathbf{h}_0\|_2, \tag{1}$$

where $\mathbf{g}_0 \sim p_0$ and $\mathbf{g}_1 \sim p_1$. In addition, given coupling $\Gamma(p_0, p_1)$, we measure the distribution similarity between two distributions denoted as $p_0$ and $p_1$ based on the probability transport cost as follows:

$$C_g = \mathbb{E}[c_g(\mathbf{g}_0, \mathbf{g}_1)], (\mathbf{g}_0, \mathbf{g}_1) \in \Gamma(p_0, p_1). \tag{2}$$

**Geometric Optimal Transport Problem.** Building upon the established transport cost, we can formulate the geometric optimal transport problem for fast and effective 3D molecule generation. In particular, a molecule remains invariant for any rotation, translation, and permutation, while the transport cost is not invariant or equivariant to the above operations. Therefore, there exists a minimum molecule transport cost with 1) optimal rotation and translation transformations such that the molecules from the data and noise are nearest to each other at the atomic coordinate level and 2) optimal permutation transformation such that the atomic features of the two are also nearest.

We supplement the detailed analysis of equivariance and invariance in Appendix A and present geometric optimal transport as follows:

$$\min_{\Gamma} \mathbb{E}[\hat{c}_g(\mathbf{g}_0, \mathbf{g}_1)],$$
$$\textbf{s. t.} \quad (\mathbf{g}_0, \mathbf{g}_1) \in \Gamma(p_0, p_1),$$
$$\hat{c}_g(\mathbf{g}_0, \mathbf{g}_1) = \lambda \min_{\mathbf{R}, \mathbf{t}, \pi} \|\pi(\mathbf{R}\mathbf{x}_1^1 + \mathbf{t}, \mathbf{R}\mathbf{x}_1^2 + \mathbf{t}, \dots, \mathbf{R}\mathbf{x}_1^N + \mathbf{t}) - (\mathbf{x}_0^1, \mathbf{x}_0^2, \dots, \mathbf{x}_0^N)\|_2 \tag{3}$$
$$+ (1 - \lambda) \min_{\pi} \|\pi(\mathbf{h}_1^1, \mathbf{h}_1^2, \dots, \mathbf{h}_1^N) - (\mathbf{h}_0^1, \mathbf{h}_0^2, \dots, \mathbf{h}_0^N)\|_2, \forall \pi, \mathbf{R}, \text{ and } \mathbf{t},$$

where $\hat{c}_g$ denotes optimal molecule transport cost, $\pi$ represents a permutation of $N$ elements, $\lambda$ is a trade-off coefficient balancing the transport costs of atom coordinates and atom features for searching optimal permutation ($\hat{\pi}$). Additionally, $\mathbf{R}$ and $\mathbf{t}$ denote a rotation matrix and a translation, respectively. The defined geometric optimal transport problem forms a bi-level optimization problem that involves two levels of optimization tasks. Specifically, minimizing molecule transport cost is nested inside the optimizing distribution transport cost.

**The Challenges of Solving the Geometric Optimal Transport Problem.** *First*, optimal molecule transport involves searching for a unified optimal permutation for atom coordinates and features with minimum transport cost. The paths for transporting continuous coordinates and categorical features are incompatible and require sophisticated, hybrid modeling of multi-modal variables (Song et al., 2023a), leading to a sub-optimal solution. *Second*, a molecular distribution comprises molecules with diverse numbers of atoms, introducing difficulties in quantifying the transport cost for searching optimal coupling. As a result, the minimization of geometric transport $C_g$ within molecular distributions poses a more significant challenge compared to other domains such as computer vision (Tong et al., 2023) or many-body systems (Klein et al., 2023). Moreover, the proposed geometric optimal transport problem, which involves a nested optimization structure, presents a significant computational challenge for optimization.

## 3.2 Solving Geometric Optimal Transport Problem

In this section, we will address the above challenges under the depicted problem for fast and effective 3D molecule generation from two aspects, including optimal molecule transport and optimal distribution transport.

### 3.2.1 Solving Optimal Molecule Transport

As illustrated in Eq. (3), our objective for optimal molecule transport is to find $\hat{\mathbf{R}}$, $\hat{\mathbf{t}}$, and $\hat{\pi}$ that minimize the transport cost denoted as $c_g$ for two given molecule geometries represented as $\mathbf{g}_0$ and $\mathbf{g}_1$:

$$\hat{\mathbf{R}}, \hat{\mathbf{t}}, \hat{\pi} = \arg \min_{\mathbf{R}, \mathbf{t}, \pi} c_g(\mathbf{g}_0, \mathbf{g}_1), \forall \pi, \mathbf{R}, \text{ and } \mathbf{t}. \tag{4}$$

As per Eq. (3), the optimal molecule transport problem entails the consideration of both atom coordinates and features for comprehensive representations of 3D molecules. Though coordinates and features represent different modalities, they need to be considered in tandem. Previous research (Song et al., 2023a; Klein et al., 2023) either solely focused on equivariant optimal transport for coordinates or transported atom features via a distinct yet non-optimal path. In contrast, we propose to unify the transport of atom coordinates and features. If we can unify different modalities within an equivariant and smooth representation space, optimal transport from a base distribution to the data distribution, leading to fast and effective molecule generation, is possible.

Specifically, we map the atom coordinates and features from the data space into a latent space with an equivariant autoencoder (Satorras et al., 2021), which enables us to compute a unified optimal permutation ($\hat{\pi}$).

After the mapping, we ascertain the optimal rotation ($\hat{\mathbf{R}}$) for two atomic coordinate sets for transportation, utilizing the Kabsch algorithm (Kabsch, 1976) as did in (Song et al., 2023a; Klein et al., 2023). The computed rotation matrix ensures that the coordinates of the target molecules in the latent space are in closest proximity to the noise molecules. In addition, to achieve optimal translation ($\hat{\mathbf{t}}$), we establish the data distribution $p_1$ and base distribution $p_0$ in the zero CoM space (Hoogeboom et al., 2022; Xu et al., 2023). Equipped with the equivariant autoencoder, the latent representation also resides in the zero CoM space, thus ensuring optimal translation in the latent space.

**Implementation.** Initially, the distributions $p_0$ and $p_1$ are aligned to the Zero CoM by subtracting the center of gravity using $\hat{\mathbf{t}}$.

Subsequently, an equivariant autoencoder is designed to project $\mathbf{g}_1 \sim p_1$ into the latent space. Here, the encoder $\mathcal{E}_\phi$ transforms $\mathbf{g}_1$ into latent domain $\mathbf{z}_1 = \mathcal{E}_\phi(\mathbf{g}_1)$, where $\mathbf{z}_1 = \langle \mathbf{z}_{\mathbf{x},1} \in \mathbb{R}^{N \times 3}, \mathbf{z}_{\mathbf{h},1} \in \mathbb{R}^{N \times k} \rangle$ and $k$ represents the latent dimensionality for the atomic features. The decoder $\mathcal{D}_\epsilon$ then learns to decode $\mathbf{z}_1$ back to molecular domain formulated as $\hat{\mathbf{g}_1} = \mathcal{D}_\epsilon(\mathbf{z}_1)$. The equivariant autoencoder can be trained by minimizing the reconstruction objective, which is $d(\mathcal{D}(\mathcal{E}(\mathbf{g}_1)), \mathbf{g}_1)$. With the encoded $\mathbf{z}_1$ and the sampled noise $\mathbf{z}_0 = \langle \mathbf{z}_{\mathbf{x},0} \in \mathbb{R}^{N \times 3}, \mathbf{z}_{\mathbf{h},0} \in \mathbb{R}^{N \times k} \rangle$ from $p_0$, we then formulate the atom-level cost matrix as $M_{c_g}[i,j] = \|\mathbf{z}_1^i - \mathbf{z}_0^j\|^2 = \|\mathbf{z}_{\mathbf{x},1}^i - \mathbf{z}_{\mathbf{x},0}^j\|^2 + \|\mathbf{z}_{\mathbf{h},1}^i - \mathbf{z}_{\mathbf{h},0}^j\|^2$, which is the 2-norm distance between $i$-th atom of $\mathbf{z}_1$ and $j$-th atom of $\mathbf{z}_0$ including the latent coordinates $\mathbf{z}_{\mathbf{x}}$ and the latent features $\mathbf{z}_{\mathbf{h}}$. With $M_{c_g}$, the optimal permutation $\hat{\pi}$ is induced with the Hungarian algorithm (Kuhn, 1955). The coordinates of the noise molecule $\mathbf{z}_{\mathbf{x},0}$ and the latent coordinates of the target molecule $\mathbf{z}_{\mathbf{x},1}$ are then aligned through rotation $\hat{\mathbf{R}}$ solved by the Kabsch algorithm (Kabsch, 1976). In summary, we perform the above-calculated translation, encoding, rotation, and permutation on the target molecule $\mathbf{g}_1$ to obtain $\hat{\mathbf{z}}_1$, which forms the optimal molecule transport cost with the sampled noise $\mathbf{z}_0$. The complete process is denoted as $\hat{\mathbf{z}}_1 = \pi(\hat{\mathbf{R}}\mathcal{E}_\phi(\mathbf{g}_1 + \hat{\mathbf{t}}))$.

### 3.2.2 Searching Optimal Coupling for Optimal Distribution Transport

By solving Eq. (4), we can obtain $\hat{\mathbf{R}}$, $\hat{\mathbf{t}}$, and $\hat{\pi}$ yielding an optimal molecule transport trajectory — a straight one — given two data points from the base distribution and target distribution, respectively (see the gray trajectory in Figure 2). Nevertheless, ensuring a straight trajectory does not necessarily yield optimal transport for generative modeling because a straight map for two data points does not indicate a straight map between two distributions. Figure 2 depicts two possible trajectories for

generative modeling. The gray one shows a straight but not the shortest map, while the red one represents the shortest map, indicating an optimal trajectory. As discussed in Sec. 3.1, an optimal trajectory can only be achieved with optimal coupling, leading to the shortest path for mapping the base distribution to the target distribution. To approximate optimal coupling and further boost the sampling speed, we introduce the second part of our framework, optimal flow estimation and purification, which is dedicated to solving optimal distribution transport.

The pathway to optimal distribution transport is to identify the optimal coupling $\hat{\Gamma}(p_0, p_1)$ that satisfies the condition formulated as:

$$\hat{\Gamma} = \arg \min_{\Gamma} \mathbb{E}[\hat{c}_g(\mathbf{z}_0, \mathbf{z}_1)]$$
$$\text{s. t. } (\mathbf{z}_0, \mathbf{z}_1) \in \Gamma(p_0, p_1), \tag{5}$$

where $\Gamma$ is an arbitrary coupling plan between $p_0$ and $p_1$ and $\hat{c}_g$ is optimal molecule transport cost defined in Eq. (3).

However, measuring the distribution transport cost for searching for optimal coupling is challenging due to the various sizes of molecules. Inspired by (Liu et al., 2022), we circumvent the need to quantify transport costs by estimating the optimal coupling, $\hat{\Gamma}(p_0, p_1')$ based on a trained flow model with the initial coupling denoted as $\Gamma(p_0, p_1)$.

The estimated optimal coupling can minimize the transport cost but may introduce generation error for the following reasons. The first type of error arises from estimating the flow path between $p_0$ to $p_1$ via a

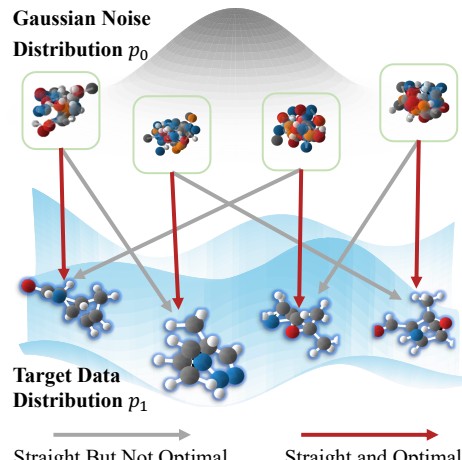

**Gaussian Noise Distribution $p_0$**

**Target Data Distribution $p_1$**

Straight But Not Optimal     Straight and Optimal

Figure 2: **An Illustration of the Difference Between Straight Coupling and Optimal Coupling.** GOAT approximates optimal coupling for a fast generation.

neural network $v_\theta$, implying that $p_1'$, characterized by $v_\theta$, does not perfectly match the data distribution $p_1$. The second type of error stems from the discreteness of molecular data and the continuity of the distribution. In essence, an intermediate value between two similar and valid molecules, which are closely distributed, may not be biochemically valid. To compensate for such discrepancies, we implement a purification process on the generated coupling to ensure effective generation. We present a detailed implementation below.

**Implementation.** First, based on Sec. 3.2.1, we can obtain a set of noise and target molecule pairs with optimal molecule transport cost. We leverage the corresponding transport path as the conditional probability path for training the flow with the loss formulated as:

$$\mathcal{L}_{F1}(\theta) = \mathbb{E}_{t,p_0,p_1} \| v_\theta(\hat{\mathbf{z}}_t, t) - (\hat{\mathbf{z}}_1 - \mathbf{z}_0) \|^2, \tag{6}$$

where $\hat{\mathbf{z}}_t = t\hat{\mathbf{z}}_1 + (1 - t)\mathbf{z}_0, t \in [0, 1]$. Compared with using conditional optimal transport path and variance-preserving path in a hybrid fashion (Song et al., 2023a), our method employs a unified linear interpolation of $\hat{\mathbf{z}}_1 - \mathbf{z}_0$ as the flow probability path. Such straight trajectory adheres to the naive ODE formula denoted as $d\mathbf{z}_t = (\hat{\mathbf{z}}_1 - \mathbf{z}_0)dt$, thereby providing a more straight flow path for fast sampling. The optimum of $\mathcal{L}_{F1}$ is achieved when $v_{\hat{\theta}}(\mathbf{z}_t, t) = \mathbb{E}[\mathbf{z}_1 - \mathbf{z}_0 | \mathbf{z}_t]$.

The proposed framework then samples data pairs $(\mathbf{z}_0, \mathbf{z}_1')$ via trained flow model $\hat{\theta}_1$ as the estimated *optimal coupling*. Specifically, $\mathbf{z}_1'$ can be sampled following $d\mathbf{z}_t = v_{\hat{\theta}_1}(\mathbf{z}_t, t)dt$ starting from $\mathbf{z}_0 \sim p_0$ and the process of sampling is denoted as $\text{ODE}_{v_{\hat{\theta}}}$. Pair of $\mathbf{z}_0$ and $\mathbf{z}_1'$ is set as fixed and a batch of pairs will be generated as the estimated optimal coupling represented as $\hat{\Gamma}(p_0, p_1') = \{(\mathbf{z}_0, \mathbf{z}_1')\}$, where $\mathbf{z}_1' = \text{ODE}_{v_{\hat{\theta}}}(\mathbf{z}_0)$. Finally, we *estimate* and *purify* the flow. Specifically, $\mathbf{z}_1'$ is decoded by $\mathcal{D}_\epsilon$ for $\mathbf{g}_1'$ and evaluated in terms of stability and validity by RdKit (Landrum et al., 2016). This provides a criterion for filtering out invalid molecules to purify the coupling. Subsequently, the optimal flow is trained using the loss in Eq. (6) with estimated and purified coupling.

**Provably Reduced Geometric Transport Cost.** The estimated optimal coupling $\hat{\Gamma}$ can boost generation only when geometric transport cost is reduced. We theoretically show that our approach can indeed reduce geometric transport costs as follows:

**Theorem 3.1.** *The coupling $\hat{\Gamma}$ incurs no larger geometric transport cost than the random coupling $\Gamma(p_0, p_1)$ in that $\mathbf{E}[\hat{c}_g(\mathbf{z}_0, \mathbf{z}_1')] \leq \mathbf{E}[\hat{c}_g(\mathbf{z}_0, \mathbf{z}_1)]$, where $(\mathbf{z}_0, \mathbf{z}_1') = \hat{\Gamma}(p_0, p_1')$, $(\mathbf{z}_0, \mathbf{z}_1) = \Gamma(p_0, p_1)$, and $\hat{c}_g$ is optimal molecule transport cost.*

With this theorem, the proposed GOAT is guaranteed a Pareto descent on the geometric transport cost, leading to faster generation. A comprehensive proof is given in Appendix B, and the pseudocode for training and sampling is presented in Appendix C.

## 4 EXPERIMENTAL STUDIES

**Datasets.** We evaluate over benchmark datasets for 3D molecule generation, including QM9 (Ramakrishnan et al., 2014) and the GEOM-DRUG (Axelrod & Gómez-Bombarelli, 2022). QM9 is a standard dataset that contains 130k 3D molecules with up to 29 atoms. GEOM-DRUG encompasses around 450K molecules, each with an average of 44 atoms and a maximum of 181 atoms. More dataset details are presented in Appendix E.

**Baselines.** We compare GOAT with several competitive baseline models. G-Schnet (Gebauer et al., 2019) and Equivariant Normalizing Flows (ENF) (Chen et al., 2018) are equivariant generative models utilizing the autoregressive models and continuous normalizing flow, respectively. Equivariant Graph Diffusion Model (EDM) and its variant GDM-Aug (Hoogeboom et al., 2022), EDM-Bridge (Wu et al., 2022), GeoLDM (Xu et al., 2023) are diffusion-based approaches. GeoBFN (Song et al., 2023b) leverages Bayesian flow networks for distributional parameter approximation. EquiFM (Song et al., 2023a) is the first flow-matching method for 3D molecule generation.

### 4.1 EVALUATION METRICS

**Evaluating Generation Quality.** Without loss of generality, we use validity, uniqueness, and novelty to evaluate the quality of generated molecules (Zhang et al., 2023). Existing experiments calculate validity, uniqueness, and novelty, which are nested; novelty measures novel molecules among unique and valid molecules. However, such a calculation cannot reflect the ultimate quality among all samples. We further propose a new metric toward the significance of generative models (Walters & Murcko, 2020).

Below, we provide the detailed definitions of these metrics. 1) **Validity.** An essential criterion for molecule generation is that the generated molecules must be chemically valid, which implies that the molecules should obey chemical bonds and valency constraints. We use RdKit (Landrum et al., 2016) to check if a molecule obeys the chemical valency rules. Validity calculates the percentage of valid molecules among all the generated molecules; 2) **Uniqueness.** An important indicator of a molecule generative model is whether it can continuously generate different samples, which is quantified by the uniqueness. We evaluate uniqueness by measuring the fraction of unique molecules among all the generated valid ones; 3) **Novelty.** An ideal generative model for *de novo* molecule design should be able to generate novel molecular samples that do not exist in the training set. Therefore, we report novelty that quantifies the percentage of novel samples among all the valid and unique molecules; 4) **Significance.** To comprehensively evaluate the molecule generative models, we represent a new metric, *significance*, to quantify the percentage of valid, unique, and novel molecules among the generated samples.

**Evaluating Generation Efficiency.** 1) We report sampling **steps** to measure the generation speed. The time cost of each sampling step in most baselines, including EDM, EDM-Bridge, GeoBFN, GeoLFM, and EquiFM, is identical because they all applied EGNN (Satorras et al., 2021) with the same layers and parameters. Fewer steps indicate higher generation efficiency. For EquiFM and the proposed GOAT, we applied the same adaptive stepsize on ODE solver Dopri5 (Dormand & Prince, 1980) for a fair comparison. 2) Considering the generation quality and efficiency simultaneously, we propose to report the m il (valid, unique, and novel) molecule, denoted as **S-Time**. The metric is calculated by the number of significant molecules over the total time consumed by generating all molecules, and it comprehensively reflects the performance of generation quality and efficiency. 3) We measure the generation efficiency by comparing geometric transport cost, which is calculated by Eq. (2).

## 4.2 RESULTS AND ANALYSIS

Table 1: Comparisons of generation quality regarding Atom Stability, Validity, Uniqueness, Novelty, and Significance. And comparisons of generation efficiency regarding Steps and Time. The **best** results are highlighted in bold.

| QM9 | Quality (↑) | | | | | Efficiency (↓) | |
|---|---|---|---|---|---|---|---|
| Metrics | Atom Sta | Valid | Uniqueness | Novelty | Significance | Steps | S-Time |
| Data | 99.0 | 97.7 | 100.0 | - | - | - | - |
| ENF | 85.0 | 40.2 | 98.0 | - | - | - | - |
| G-Schnet | 95.7 | 85.5 | 93.9 | - | - | - | - |
| GDM-aug | 97.6 | 90.4 | **99.0** | 74.6 | 66.8 | 1000 | 1.50 |
| EDM | 98.7 | 91.9 | 98.7 | 65.7 | 59.6 | 1000 | 1.68 |
| EDM-Bridge | 98.8 | 92.0 | 98.6 | - | - | 1000 | - |
| GeoLDM | 98.9 | 93.8 | 98.8 | 58.1 | 53.9 | 1000 | 1.86 |
| GeoBFN | 98.6 | 93.0 | 98.4 | 70.3 | 64.4 | 100 | 0.16 |
| EquiFM | 98.9 | **94.7** | 98.7 | 57.4 | 53.7 | 200 | 0.37 |
| **GOAT (Ours)** | **99.2** | 92.9 | **99.0** | **78.6** | **72.3** | **90** | **0.12** |

Table 2: Comparisons of generation quality regarding Atom Stability, Validity, Steps, and Time on GEOM-DRUG. The **best** results are highlighted in bold.

| GEOM-DRUG | Quality (↑) | | Efficiency (↓) | |
|---|---|---|---|---|
| Metrics | Atom Sta | Valid | Steps | S-Time |
| Data | 86.5 | 99.9 | - | - |
| GDM-aug | 77.7 | 91.8 | 1000 | - |
| EDM | 81.3 | 92.6 | 1000 | 14.88 |
| EDM-Bridge | 82.4 | 92.8 | 1000 | - |
| GeoLDM | 84.4 | **99.3** | 1000 | 12.84 |
| GeoBFN | 78.9 | 93.1 | 100 | 1.27 |
| EquiFM | 84.1 | 98.9 | 200 | 2.02 |
| **GOAT (Ours)** | **84.8** | 96.2 | **90** | **0.94** |

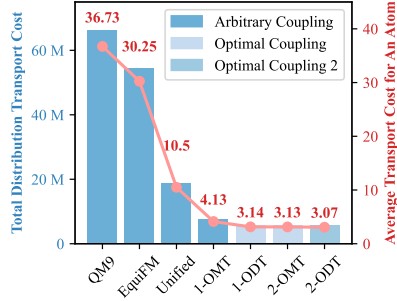

Figure 3: The blue histogram plots the comparisons of distribution transport cost. The red line chart depicts the average transport cost per atom (best view in color).

In this study, we generate 10K molecular samples for each method and compute the aforementioned metrics for comparisons. The evaluation results are presented in Tables 1 and 2 with Figure 3.

**Performance Comparisons with Diffusion-Based Methods.** We observe that all diffusion-based generation methods indeed need 1000 sampling steps to achieve comparable generation quality. Surprisingly, with the least sampling steps, GOAT achieves the best atom stability, uniqueness, novelty, and significance over QM9. Specifically, it improves novelty by up to 35.2% and significance by up to 34.1%, respectively. Among these diffusion models, GeoLDM achieves the best validity performance. However, it owns relatively poor novelty and significance, 58.1% and 53.9% on QM9, respectively. These results indicate that the latent diffusion models can model the complex geometric 3D molecules well but introduce a serious overfitting problem — generating more molecules that are the same as the training samples. Though GDM-Aug can achieve the second-best novelty among all methods, it needs 1000 sampling steps for 3D molecule generation. As for GEOM-DRUG, we directly compare the validity as ultimate significance since all compared methods achieved almost 100% uniqueness (Xu et al., 2023). Table 2 shows that the proposed algorithm also achieves competitive performance while maintaining a leading edge in generation speed on such a large-scale dataset. Specifically, GOAT only spends 0.94 seconds for each valid, unique, and novel molecule on average and reaches 96.2% validity, while GeoLDM takes more than 10× seconds to reach 99.3%. We believe this performance is competitive and more efficient.

**Performance Comparisons with Flow-Matching-Based Methods.** EquiFM and GOAT are all based on flow matching, using an ODE solver for generation. We can see that flow-matching-based methods can obtain faster generation speeds than diffusion models. In particular, GOAT only needs 90 steps, while EquiFM requires 200 steps for sampling. EquiFM solely considers optimal transport for atom coordinates. Therefore, the generation speed is still inferior to ours. Because the proposed GOAT solves optimal molecule transport and optimal distribution transport together, the number

Table 3: MAE for molecular property prediction. A lower number indicates a better controllable generation result. The **best** results are highlighted in bold.

| Property | Steps | $\alpha$ | $\Delta\varepsilon$ | $\varepsilon_{\text{HOMO}}$ | $\varepsilon_{\text{LUMO}}$ | $\mu$ | $C_v$ |
|---|---|---|---|---|---|---|---|
| Units | | Bohr$^3$ | meV | meV | meV | D | $\frac{\text{cal}}{\text{mol}}$ K |
| QM9 | - | 0.100 | 64 | 39 | 36 | 0.043 | 0.040 |
| Random | - | 9.010 | 1470 | 645 | 1457 | 1.616 | 6.857 |
| N atoms | - | 3.860 | 866 | 426 | 813 | 1.053 | 1.971 |
| EDM | 1000 | 2.760 | 655 | 356 | 583 | 1.111 | 1.101 |
| GeoLDM | 1000 | 2.370 | 587 | 340 | 522 | 1.108 | 1.025 |
| EquiFM | 220 | 2.410 | 591 | 337 | 530 | 1.106 | 1.033 |
| **GOAT (Ours)** | **200** | **1.725** | **585** | **330** | **521** | **0.906** | **0.881** |
| GeoBFN | 100 | 3.875 | 768 | 426 | 855 | 1.331 | 1.401 |
| EquiFM | 100 | 3.006 | 830 | 392 | 735 | 1.064 | 1.177 |
| **GOAT (Ours)** | **100** | **2.740** | **605** | **350** | **534** | **1.010** | **0.883** |

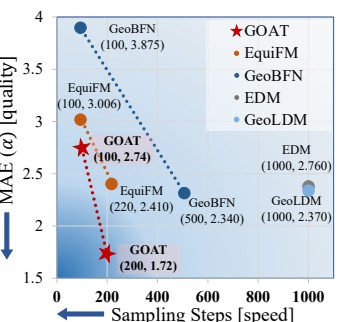

Figure 4: **Quality vs. Speed ($\alpha$).** GOAT shows the optimal trade-off between generation quality and speed.

of sampling steps is further reduced by 2× compared to EquiFM with the same ODE solver. This verifies our hypothesis that a joint optimal transport path can further boost the generation efficiency.

Though EquiFM can perform well in terms of molecule validity, it achieves unsatisfactory performance in novelty and significance on QM9 among all methods. More specifically, nearly half of the generated samples are the same as the training samples, which is unacceptable in the context of *de novo* molecule design. In contrast, GOAT can obtain 78.6% novelty with 37% improvement and 72.3% significance with 34.6% improvement compared to EquiFM. On GEOM-DRUG, the proposed method achieves approximate performance compared to EquiFM while taking only half the sampling steps. GeoBFN (Song et al., 2023b) can have comparable sampling efficiency to ours, which is neither diffusion-based nor flow-matching based methods. We find that its generation quality over GEOM-DRUG is around 3% lower than GOAT regarding the validity, and it owns around 8% decrease in novelty with a similar sampling speed.

**Geometric Transport Cost Comparisons.** As EquiFM and GOAT are both flow-matching-based transport methods, we compare their transport costs and present the visualized results in Figure 3. We present distribution transport cost ($p_0 \rightarrow p_1$) in blue bars and molecule transport cost averaged over the number of atoms ($\mathbf{g}_0 \rightarrow \mathbf{g}_1$) in red lines. Compared to EquiFM transports with a hybrid method, the proposed method reduced the geometric transport cost with 1) unified transport (Unified), 2) optimal molecule transport (1-OMT), and 3) optimal distribution transport cost (1-ODT), thereby achieving a significant reduction in geometric transport cost by nearly 89.65%, leading to faster generation. We further minimize molecule and distribution transport costs (2-OMT and 2-ODT) and observe that the transport cost is reduced marginally, indicating a nearly optimum of the proposed method. The above analysis reveals that the proposed method indeed reduced the geometric transport cost by unifying transport, minimizing molecule transport cost, and estimating optimal couplings. The most intuitive manifestation of the reduction in transport cost is the boost in generation speed, which has been demonstrated in the previous section.

**Controllable Molecule Generation.** Without loss of generality, GOAT can be readily adapted to perform controllable molecule generation with a desired property $s$ by modeling the neural network as $v_\theta(\mathbf{z}, t|s)$. We evaluated the performance of GOAT on generating molecules with properties including $\alpha$, $\Delta\varepsilon$, $\varepsilon_{\text{HOMO}}$, $\varepsilon_{\text{LUMO}}$, $\mu$, and $C_v$. The quality of the generated molecules concerning their desired property was assessed using the Mean Absolute Error (MAE) between the conditioned property and the predicted property. This measure helps to determine how closely the generated molecules align with the desired property.

We use the property classifier network $\varphi$ from (Garcia Satorras et al., 2021) and split the QM9 training partition into two halves with 50K samples each. The classifier $\varphi$ is trained in the first half, while the Conditional GOAT is trained in the second half. Then, $\varphi$ is applied to evaluate conditionally generated samples by the GOAT. We report the numerical results in Table 3. Random means we simply do random shuffling of the property labels in the dataset and then evaluate $\varphi$ on it. $N_{\text{atoms}}$ predicts the molecular properties by only using the number of atoms in the molecule.

Compared to existing methods, our proposed approach demonstrates superior performance in controllable generation tasks. Specifically, it achieves the best results across all six tasks when evaluated with 100 sampling steps, outperforming other variable sampling step methods (EquiFM and GeoBFN). Furthermore, even with increased sampling steps, our method maintains outstanding per-

Table 4: Ablation Studies. OMT represents optimal molecule transport, and ODT stands for optimal distribution transport. The **best** results are highlighted in bold, and the second-best results are highlighted with underlines.

| Metrics (QM9) | | Quality (↑) | | | | | Efficiency (↓) | | |
|---|---|---|---|---|---|---|---|---|---|
| Components | $\lambda$ | Atom Sta | Valid | Uniqueness | Novelty | Significance | Steps | Time | Cost |
| w/o EAE | - | 98.8 | 92.8 | 92.2 | 58.4 | 53.9 | 280 | 0.63 | 30.25 |
| w/o ODT | 1 | 97.7 | 89.5 | 98.7 | 70.1 | 61.9 | 120 | 0.17 | 5.01 |
| w/o ODT | 0.75 | 97.9 | 89.7 | 98.9 | 70.2 | 62.3 | 110 | 0.16 | 4.63 |
| w/o ODT | 0.5 | 98.1 | 89.9 | 98.8 | 70.4 | 62.5 | 100 | 0.14 | 4.13 |
| w/o ODT | 0.25 | 97.8 | 89.6 | 98.7 | 70.1 | 62.0 | 120 | 0.17 | 4.87 |
| w/o ODT | 0 | 97.5 | 89.3 | 98.8 | 70.0 | 61.8 | 130 | 0.19 | 5.41 |
| w/o OMT | 0.5 | 96.5 | 85.0 | 98.9 | 69.1 | 58.1 | 170 | 0.31 | 5.32 |
| GOAT | 0.5 | **99.2** | **92.9** | **99.0** | **78.6** | 72.3 | **90.0** | **0.12** | **3.14** |

formance in generating molecules with properties $\alpha$, $\mu$, and $C_v$, whereas other methods require longer sampling steps to achieve comparable results. Notably, only GeoBFN, which requires more than doubled sampling steps, shows a marginal advantage in other properties. To better illustrate the advantages of the proposed method, we present a performance comparison in accuracy and efficiency, as measured by property $\alpha$, in Figure 4. The figure demonstrates that the proposed method achieves a new trade-off between accuracy and efficiency in conditional molecule generation.

**Parameter Analysis and Ablation Studies.** In this section, we first analyze the effectiveness of the parameter $\lambda$, which determines the trade-off between transporting atom coordinates and atom features. Since the parameter only affects the optimization of molecule transport, we compare its influence without considering optimal distribution transport. Our results in Table 4 indicate that when $\lambda = 0.5$, meaning the weights for optimizing transport costs in two modalities are equal, the proposed algorithm achieves the best transport plan. We hypothesize that this is because the transport cost calculated by equation 3 with $\lambda = 0.5$ accurately reflects the actual cost for the generative model in transporting noise to the data distribution.

Additionally, we conduct ablation studies on equivariant autoencoder (EAE), optimal molecule transport, and optimal distribution transport. Without considering optimal transport, the model trained solely with flow matching in the latent space (w/o OMT) shows a significant increase in training speed. This can be attributed to the reduced transport cost resulting from the unified space, although the performance remains suboptimal. When solving OMT without ODT (w/o ODT, $\lambda = 0.5$), both performance and speed improve, but they still do not reach the final results, which account for both molecule and distribution in geometric optimal transport.

**Limitations.** Addressing the optimal transport costs, particularly those involving rotation and permutation aspects, can be computationally intensive (Song et al., 2023a; Klein et al., 2023). However, these operations can be efficiently parallelized on CPUs to enhance the training speed. Besides, refining the flow may require additional time-consuming training, but such an operation boosts the generation speed and improves novelty without compromising quality. In summary, the above-mentioned operations will accelerate the generation of molecules once and for all after training, which is prioritized in this research. We leave improvements concerning training efficiency and other methods for boosting generation speed, such as distillation (Liu et al., 2022), for future work.

## 5  CONCLUSION

This paper introduces GOAT, a 3D molecular generation framework that tackles optimal transport for enhanced generation quality and efficiency in molecule design. Recognizing that in silico molecule generation is a problem of probability distribution transport and the key to accelerating this lies in minimizing the transport cost. To this end, we formulated the geometric optimal transport problem tailored for molecular distribution. This proposed problem led us to consider the transport cost of atom coordinates, atom features, and the complete molecules. This motivates the design of joint transport to solve optimal molecule transport with different modalities and the framework to minimize the distributional transport cost. Both theoretical and empirical validations confirm that GOAT reduces the geometric transport cost, resulting in faster and more effective molecule generation. Our method achieves state-of-the-art performance in generating valid, unique, and novel molecules, thereby enhancing the ultimate significance of in-silico molecule generation.

ACKNOWLEDGMENTS

This work was supported in part by the Research Grants Council of the Hong Kong (HK) SAR under Grant No. C5052-23G, Grant PolyU 15229824, Grant PolyU 15218622, Grant PolyU 15215623 and Grant PolyU 15208222; the National Natural Science Foundation of China (NSFC) under Grants U21A20512; NSFC Young Scientist Fund under Grant PolyU A0040473.

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

APPENDIX

## A EQUIVARIANCE AND INVARIANCE IN GEOMETRIC OPTIMAL TRANSPORT

**Equivariance.** Molecules, typically existing within a three-dimensional physical space, are subject to geometric symmetries, including translations, rotations, and potential reflections. These are collectively referred to as the Euclidean group in 3 dimensions, denoted as E(3) (Celeghini et al., 1991).

A function $F$ is said to be equivariant to the action of a group $G$ if $T_g \circ F(\mathbf{x}) = F \circ S_g(\mathbf{x})$ for all $g \in G$, where $S_g, T_g$ are linear representations related to the group element $g$ (Serre et al., 1977).
**Invariance.** A function $F$ is said to be invariant to the action of a group $G$ if $F \circ \pi(\mathbf{x}) = F(\mathbf{x})$ for all $g \in G$ and every permutation $\pi \in S_n$.

**Equivariance and Invariance in Molecules.** For geometric graph generation, we consider the special Euclidean group SE(3), involving translations and rotations. Moreover, the transformations $S_g$ or $T_g$ can be represented by a translation $\mathbf{t}$ and an orthogonal matrix rotation $\mathbf{R}$.

For a molecule $\mathbf{g} = \langle \mathbf{x}, \mathbf{h} \rangle$, the node features $\mathbf{h}$ are SE(3)-invariant while the coordinates $\mathbf{x}$ are SE(3)-equivariant, which can be expressed as $\mathbf{Rx} + \mathbf{t} = (\mathbf{Rx}_1 + \mathbf{t}, \ldots, \mathbf{Rx}_N + \mathbf{t})$.

**Equivariance and Invariance in Geometric Optimal Transport.** For non-topological data, such as images, the transport cost between two given data points is fixed. However, this does not apply to topological graphs. For instance, when a topological graph (molecule) undergoes rotation or translation, the inherent properties of the molecule remain unchanged, but the cost of transporting coordinates may vary. Similarly, if the atom order in one of the molecules changes in silico, the molecule remains constant, but the transport cost of coordinates and features may alter. Therefore, the proposed optimal molecule transport problem aims to find an optimal rotation, translation, and permutation transformation for one molecule to minimize the distance, considering both coordinates and features, from another molecule.

## B PROOF FOR THEOREM 3.1

The theorem 3.1 is reproduced here for convenience:

**Theorem 3.1** *The coupling $\hat{\Gamma}$ incurs no larger geometric transport costs than the arbitrary coupling $\Gamma(p_0, p_1)$ in that $\mathbf{E}[\hat{c}_g(\mathbf{z}_0, \mathbf{z}_1')] \leq \mathbf{E}[\hat{c}_g(\mathbf{z}_0, \mathbf{z}_1)]$ where $(\mathbf{z}_0, \mathbf{z}_1') \in \hat{\Gamma}(p_0, p_1')$, $(\mathbf{z}_0, \mathbf{z}_1) \in \Gamma(p_0, p_1)$, and $\hat{c}_g(\mathbf{z}_0, \mathbf{z}_1) = \min \|\pi(\mathbf{Rz}_{\mathbf{x},1}^1 + \mathbf{t}, \mathbf{Rz}_{\mathbf{x},1}^2 + \mathbf{t}, \ldots, \mathbf{Rz}_{\mathbf{x},1}^N + \mathbf{t}) - (\mathbf{z}_{\mathbf{x},0}^1, \mathbf{z}_{\mathbf{x},0}^2, \ldots, \mathbf{z}_{\mathbf{x},0}^N)\|_2 + \min \|\pi(\mathbf{z}_{\mathbf{h},1}^1, \mathbf{z}_{\mathbf{h},1}^2, \ldots, \mathbf{z}_{\mathbf{h},1}^N) - (\mathbf{z}_{\mathbf{h},0}^1, \mathbf{z}_{\mathbf{h},0}^2, \ldots, \mathbf{z}_{\mathbf{h},0}^N)\|_2, \forall \pi, \mathbf{R},$ and $\mathbf{t}$.*

$\mathbf{z}$ is geometry $\mathbf{g}$ in the latent space, which is composed of $\mathbf{z}_\mathbf{x} \in \mathbb{R}^{N \times 3}$ and $\mathbf{z}_\mathbf{h} \in \mathbb{R}^{N \times k}$, where $k$ is the latent dimension characterized by $\mathcal{E}_\phi$.

With node-granular optimal transport $\hat{\mathbf{R}}, \hat{\mathbf{t}}$ and $\hat{\pi}$ we have:

$$
\begin{aligned}
\mathbb{E}[\hat{c}_g(\mathbf{z}_0, \mathbf{z}_1')] = {} & \mathbb{E}[\min \|\pi(\mathbf{Rz}_{\mathbf{x},1}'^1 + \mathbf{t}, \mathbf{Rz}_{\mathbf{x},1}'^2 + \mathbf{t}, \ldots, \mathbf{Rz}_{\mathbf{x},1}'^N + \mathbf{t}) - (\mathbf{z}_{\mathbf{x},0}^1, \mathbf{z}_{\mathbf{x},0}^2, \ldots, \mathbf{z}_{\mathbf{x},0}^N)\|_2 \\
& + \min \|\pi(\mathbf{z}_{\mathbf{h},1}'^1, \mathbf{z}_{\mathbf{h},1}'^2, \ldots, \mathbf{z}_{\mathbf{h},1}'^N) - (\mathbf{z}_{\mathbf{h},0}^1, \mathbf{z}_{\mathbf{h},0}^2, \ldots, \mathbf{z}_{\mathbf{h},0}^N)\|_2, \forall \pi, \mathbf{R}, \text{ and } \mathbf{t}] \\
= {} & \mathbb{E}[\|\hat{\pi}(\hat{\mathbf{R}}\mathbf{z}_{\mathbf{x},1}'^1 + \hat{\mathbf{t}}, \hat{\mathbf{R}}\mathbf{z}_{\mathbf{x},1}'^2 + \hat{\mathbf{t}}, \ldots, \hat{\mathbf{R}}\mathbf{z}_{\mathbf{x},1}'^N + \hat{\mathbf{t}}) - (\mathbf{z}_{\mathbf{x},0}^1, \mathbf{z}_{\mathbf{x},0}^2, \ldots, \mathbf{z}_{\mathbf{x},0}^N)\|_2 \\
& + \|\hat{\pi}(\mathbf{z}_{\mathbf{h},1}'^1, \mathbf{z}_{\mathbf{h},1}'^2, \ldots, \mathbf{z}_{\mathbf{h},1}'^N) - (\mathbf{z}_{\mathbf{h},0}^1, \mathbf{z}_{\mathbf{h},0}^2, \ldots, \mathbf{z}_{\mathbf{h},0}^N)\|_2]
\end{aligned}
$$

Let $\hat{\mathbf{z}}_{\mathbf{x}} = \hat{\pi}(\hat{\mathbf{R}}\mathbf{z}_{\mathbf{x}}^1 + \hat{\mathbf{t}}, \hat{\mathbf{R}}\mathbf{z}_{\mathbf{x}}^2 + \hat{\mathbf{t}}, \ldots, \hat{\mathbf{R}}\mathbf{z}_{\mathbf{x}}^N + \hat{\mathbf{t}})$, $\hat{\mathbf{z}}_{\mathbf{h}} = \hat{\pi}(\mathbf{z}_{\mathbf{h}}^1, \mathbf{z}_{\mathbf{h}}^2, \ldots, \mathbf{z}_{\mathbf{h}}^N)$, and $\hat{\mathbf{z}} = [\hat{\mathbf{z}_x}, \hat{\mathbf{z}_h}] \in \mathbb{R}^{N \times (3+k)}$, then we have:

$$
\begin{aligned}
\mathbb{E}[\hat{c}_g(\mathbf{z}_0, \mathbf{z}_1')] =& \mathbb{E}[\|(\hat{\mathbf{z}}_{\mathbf{x},1}'^1 +, \hat{\mathbf{z}}_{\mathbf{x},1}'^2 +, \ldots, \hat{\mathbf{z}}_{\mathbf{x},1}'^N) - (\mathbf{z}_{\mathbf{x},0}^1, \mathbf{z}_{\mathbf{x},0}^2, \ldots, \mathbf{z}_{\mathbf{x},0}^N)\|_2 \\
& + \|(\hat{\mathbf{z}}_{\mathbf{h},1}'^1, \hat{\mathbf{z}}_{\mathbf{h},1}'^2, \ldots, \hat{\mathbf{z}}_{\mathbf{h},1}'^N) - (\mathbf{z}_{\mathbf{h},0}^1, \mathbf{z}_{\mathbf{h},0}^2, \ldots, \mathbf{z}_{\mathbf{h},0}^N)\|_2] \\
=& \mathbb{E}[\|(\hat{\mathbf{z}}_1'^1 +, \hat{\mathbf{z}}_1'^2 +, \ldots, \hat{\mathbf{z}}_1'^N) - (\mathbf{z}_0^1, \mathbf{z}_0^2, \ldots, \mathbf{z}_0^N)\|_2] \\
=& \mathbb{E}[\|\hat{\mathbf{z}}_1' - \mathbf{z}_0\|_2].
\end{aligned}
$$

Likewise, we have:

$$
\mathbb{E}[\hat{c}_g(\mathbf{z}_0, \mathbf{z}_1)] = \mathbb{E}[\|\hat{\mathbf{z}}_1 - \mathbf{z}_0\|_2]. \tag{7}
$$

At this point, what we aim to prove is simplified to:

$$
\mathbb{E}[\|\hat{\mathbf{z}}_1' - \mathbf{z}_0\|_2] \le \mathbb{E}[\|\hat{\mathbf{z}}_1 - \mathbf{z}_0\|_2] \tag{8}
$$

*Proof.* Given that $\mathbf{z}_1' = \text{ODE}_{\hat{\theta}}(\mathbf{z}_0)$, $d\mathbf{z}_t = v_{\hat{\theta}}(\mathbf{z}_t, t)dt$, we have:

$$
\mathbb{E}[\hat{c}_g(\mathbf{z}_0, \mathbf{z}_1')] = \mathbb{E}\left[\left\|\int_0^1 v_{\hat{\theta}}(\mathbf{z}_t, t)dt\right\|_2\right] \tag{9}
$$

$\|\cdot\|_2 : \mathbb{R}^{N \times (3+k)} \to \mathbb{R}_+$ is the Euclidean norm of $\cdot$ and it is convex, therefore, with $\|\int_\Omega v dt\| \le \int_\Omega \|v\| \, dt$ induced by Jensen's inequality we have:

$$
\mathbb{E}[\hat{c}_g(\mathbf{z}_0, \mathbf{z}_1')] \le \mathbb{E}\left[\int_0^1 \left\|v_{\hat{\theta}}(\mathbf{z}_t, t)\right\|_2 dt\right]. \tag{10}
$$

With defined $v_{\hat{\theta}}(\mathbf{z}_t, t) = \mathbb{E}[\mathbf{z}_1 - \mathbf{z}_0 | \mathbf{z}_t]$, we then have:

$$
\mathbb{E}[\hat{c}_g(\mathbf{z}_0, \mathbf{z}_1')] = \mathbb{E}\left[\int_0^1 \left\|\mathbb{E}[\mathbf{z}_1 - \mathbf{z}_0 | \mathbf{z}_t]\right\|_2 dt\right]. \tag{11}
$$

Again, with the finite form of Jensen's inequality, we have:

$$
\begin{aligned}
\mathbb{E}[\hat{c}_g(\mathbf{z}_0, \mathbf{z}_1')] \le& \mathbb{E}\left[\int_0^1 \mathbb{E}[\|\mathbf{z}_1 - \mathbf{z}_0\|_2 | \mathbf{z}_t]dt\right] && \text{// Jensen's inequality} \\
=& \int_0^1 \mathbb{E}\left[\mathbb{E}[\|\mathbf{z}_1 - \mathbf{z}_0\|_2 | \mathbf{z}_t]\right]dt \\
=& \int_0^1 \mathbb{E}[\|\mathbf{z}_1 - \mathbf{z}_0\|_2]dt && \text{// } \mathbb{E}[\|\mathbf{z}_1 - \mathbf{z}_0\|_2 | \mathbf{z}_t] = \|\mathbf{z}_1 - \mathbf{z}_0\|_2 \\
=& \mathbb{E}[\|\hat{\mathbf{z}}_1 - \hat{\mathbf{z}}_0\|_2] \\
=& \mathbb{E}[\hat{c}_g(\mathbf{z}_0, \mathbf{z}_1)] && \text{// By Eq. 7}
\end{aligned} \tag{12}
$$

Combining equations 9 to 12, Eq. 8 is proved. $\square$

It is important to note that solving the geometric optimal transport problem in the latent space does not necessarily ensure that the molecule itself or its distribution also satisfies the optimal transport in the original space. However, given that the proposed flow model is trained in the latent space, it is sufficient to ensure that latent molecules and distributions are transported with optimal cost, thereby accelerating the flow model in the generation of molecules.

## C  ALGORITHMS

This section contains the main algorithms of the proposed GOAT. First, we present the algorithm for solving optimal molecule transport and unified flow in Algorithm 1 and Algorithm 2, respectively. Algorithm 3 presents the pseudo-code for training the GOAT. Algorithm 4 presents the process of fast molecule generation with GOAT.

---

**Algorithm 1** Optimal Molecule Transport

---

1: **Input:** $\mathbf{z}_1$ and $\mathbf{z}_0$.
2: **Output:** $\hat{\mathbf{z}}_1$ and $\mathbf{z}_0$.
3: **Optimal Molecule Transport:**
4: $M_{c_g}[i,j] \leftarrow \|\mathbf{z}_1^i - \mathbf{z}_0^j\|^2 \leftarrow \|\mathbf{z}_{\mathbf{x},1}^i - \mathbf{z}_{\mathbf{x},0}^j\|^2 + \|\mathbf{z}_{\mathbf{h},1}^i - \mathbf{z}_{\mathbf{h},0}^j\|^2$ *// Construct Atom-level Transport Cost Matrix*
5: $\hat{\pi} \leftarrow$ Hungarian algorithm (Kuhn, 1955)      *// Optimal Permutation*
6: $\hat{\mathbf{R}} \leftarrow$ Kabsch algorithm (Kabsch, 1976)      *// Optimal Rotation*
7: $\hat{\mathbf{z}}_1 = \pi(\hat{\mathbf{R}}\mathbf{z}_1)$      *// Optimal Molecule Transport*
8: **return** $\hat{\mathbf{z}}_1, \mathbf{z}_0$

---

---

**Algorithm 2** Equivariant Autoencoder

---

1: **Input:** geometric data point $\mathbf{g} = \langle \mathbf{x}, \mathbf{h} \rangle$, equivariant encoder $\mathcal{E}_\phi$
2: **Output:** encoded data point $\mathbf{z}$
3: **Unified Flow:**
4: $\mathbf{x} \leftarrow \mathbf{x} - \mathbf{G}(\mathbf{x})$      *// Translate to CoM Space*
5: $\boldsymbol{\mu_x}, \boldsymbol{\mu_h} \leftarrow \mathcal{E}_\phi(\mathbf{x}, \mathbf{h})$      *// Encode*
6: $\langle \boldsymbol{\epsilon_x}, \boldsymbol{\epsilon_h} \rangle \sim \mathcal{N}(\mathbf{0}, \mathbf{I})$      *// Sample noise for Equivariant Autoencoder*
7: $\boldsymbol{\epsilon_x} \leftarrow \boldsymbol{\epsilon_x} - \mathbf{G}(\boldsymbol{\epsilon_x})$      *// Translate to CoM Space*
8: $\mathbf{z}_\mathbf{x}, \mathbf{z}_\mathbf{h} \leftarrow \mu + \langle \boldsymbol{\epsilon_x}, \boldsymbol{\epsilon_h} \rangle \odot \sigma_0$      *// Obtain Latent Representation*
9: $\mathbf{z} \leftarrow [\mathbf{z}_\mathbf{x}, \mathbf{z}_\mathbf{h}]$
10: **return** $\mathbf{z}$

---

---

**Algorithm 3** Geometric Optimal Transport

---

1: **Input:** data distribution $p_1$, equivariant encoder $\mathcal{E}_\phi$, decoder $\mathcal{D}_\epsilon$, flow network $v_\theta$
2: **Output:** GOAT: $(\hat{v}_\theta)$
3: **for** $\mathbf{g}_1 = \langle \mathbf{x}, \mathbf{h} \rangle \sim p_1$ **do**
4:      $\mathbf{z}_1 \leftarrow$ **Equivariant Autoencoder**($\mathbf{g}_1$)      *// Algorithm2*
5:      $\mathbf{z}_0 \leftarrow \langle \mathbf{z}_{\mathbf{x},0}, \mathbf{z}_{\mathbf{h},0} \rangle \sim \mathcal{N}(\mathbf{0}, \mathbf{I})$      *// Sample noise from base distribution $p_0$*
6:      $\hat{\mathbf{z}}_1, \mathbf{z}_0 =$ **Optimal Molecule Transport** ($\mathbf{z}_1, \mathbf{z}_0$)      *// Algorithm 1*
7:      $\mathcal{L}_{F1}(\theta) = \mathbb{E}_{t,p_0,p_1} \|v_\theta(\hat{\mathbf{z}}_t, t) - (\hat{\mathbf{z}}_1 - \mathbf{z}_0)\|^2$      *// Loss for the flow*
8:      $\hat{\theta} \leftarrow$ optimizer($\mathcal{L}_F, \theta$)      *// Optimize*
9: **end for**
10: **for** $\mathbf{g}_1 = \langle \mathbf{x}, \mathbf{h} \rangle \sim p_1$ **do**
11:      $\mathbf{z}_0, \mathbf{z}_1', \mathbf{g}_1' \leftarrow$ **Sampling**($\mathcal{D}_\epsilon, \hat{\theta}$)      *// Algorithm 4*
12:      **if** $\mathbf{g}_1'$ meets quality (measure by RdKit (Landrum et al., 2016)) **then**
13:          $\hat{\mathbf{z}}_1', \mathbf{z}_0 =$ **Optimal Molecule Transport** ($\mathbf{z}_1', \mathbf{z}_0$)      *// Algorithm 1*
14:          $\mathcal{L}_{F1}(\theta) = \mathbb{E}_{t,p_0,p_1} \|v_\theta(\hat{\mathbf{z}}_t', t) - (\hat{\mathbf{z}}_1' - \mathbf{z}_0)\|^2$      *// Loss for the flow*
15:          $\hat{\theta} \leftarrow$ optimizer($\mathcal{L}_F, \theta$)      *// Optimize*
16:      **end if**
17: **end for**
18: **return** $\hat{\theta}$

---

---

**Algorithm 4** Sampling

---

1: **Input:** equivariant decoder $\mathcal{D}_\epsilon$, flow network $\theta$.
2: **Output:** noise: $\mathbf{z}_0$, generated latent sample: $\mathbf{z}_1'$, generated molecule: $\mathbf{g}_1'$.
3: $\mathbf{z}_0 \leftarrow \langle \mathbf{z}_{\mathbf{x},0}, \mathbf{z}_{\mathbf{h},0} \rangle \sim \mathcal{N}(\mathbf{0}, \mathbf{I})$      *// Sample noise from base distribution $p_0$*
4: $\mathbf{z}_1' \leftarrow \text{ODE}_{v_{\hat{\theta}}}(\mathbf{z}_0)$
5: $\mathbf{g}_1' \leftarrow \mathcal{D}_\epsilon(\mathbf{z}_1')$      *// Solve ODE*
6: **return** $\mathbf{z}_0, \mathbf{z}_1', \mathbf{g}_1'$

---

## D  RELATED WORKS

**Molecule Generation Models.** Initial research in molecule generation primarily concentrated on the creation of molecules as 2D graphs (Jin et al., 2018; Liu et al., 2018; Shi et al., 2019). However, the field has seen a shift in interest towards 3D molecule generation. Techniques such as G-SchNet (Gebauer et al., 2019) and G-SphereNet (Luo & Ji, 2022) employ autoregressive methods to incrementally construct molecules by progressively linking atoms or molecular fragments. These approaches necessitate either a detailed formulation of a complex action space or an ordering of actions.

Motivated by the success of Diffusion Models (DMs) in image generation, the focus has now turned to their application in 3D molecule generation from noise (Hoogeboom et al., 2022; Xu et al., 2023; Wu et al., 2022; Han et al., 2023). To address the inconsistency of unified Gaussian diffusion across various modalities, a latent space was introduced by (Xu et al., 2023). To resolve the atom-bond inconsistency issue, (Peng et al., 2023) proposed different noise schedulers for different modalities to accommodate noise sensitivity. However, diffusion-based models consistently face the challenge of slow sampling speed, resulting in a significant computational burden for generation. To enhance the speed, recent proposals have introduced flow matching-based (Song et al., 2023a) and Bayesian flow network-based (Song et al., 2023b) models. Despite these advancements, there remains substantial potential for improvement in these frameworks regarding speed, novelty, and ultimate significance.

**Flow Models.** Introduced in (Chen et al., 2018), Continuous Normalizing Flows (CNFs) represent a continuous-time variant of Normalizing Flows (Rezende & Mohamed, 2015). Subsequently, flow matching (Lipman et al., 2022) and rectified flow (Liu et al., 2022) were proposed to circumvent the need for ODE simulations during forward and backward propagation in CNF, and they introduced optimal transport for faster generation. Leveraging these advanced flow models, (Garcia Satorras et al., 2021) pioneered the use of flow models for molecule generation, which was later followed by the proposal of (Song et al., 2023a), based on hybrid transport. Beyond the realm of 3D molecule generation, the concept of flow matching and optimal transport has also found applications in many-body systems (Garcia Satorras et al., 2021) and molecule simulations (Midgley et al., 2023). Despite these advancements, existing models primarily focus on atomic coordinates, leaving the challenge of geometric optimal transport unresolved.

## E  DATASET

### E.1  *QM9* DATASET

QM9 (Ramakrishnan et al., 2014) is a comprehensive dataset that provides geometric, energetic, electronic, and thermodynamic properties for a subset of the GDB-17 database (Ruddigkeit et al., 2012) comprises a total of 130,831 molecules [3]. We utilize the train/validation/test partitions delineated in (Anderson et al., 2019), comprising 100K, 18K, and 13K samples for each respective partition.

### E.2  *GEOM-DRUG* DATASET

*GEOM-DRUG* (Geometric Ensemble Of Molecules) dataset (Axelrod & Gómez-Bombarelli, 2022) encompasses around 450,000 molecules, each with an average of 44.2 atoms and a maximum of 181 atoms[4]. We build the GEOM-DRUG dataset following (Hoogeboom et al., 2022) with the provided code.

## F  IMPLEMENTATION DETAILS

In this study, all the neural networks utilized for the encoder, flow network, and decoder are implemented using EGNNs (Satorras et al., 2021). The dimension of latent invariant features, denoted as $k$, is set to 2 for QM9 and 1 for GEOM-DRUG, to map the molecule for a unified flow matching.

---

[3] https://springernature.figshare.com/ndownloader/files/3195389
[4] https://dataverse.harvard.edu/file.xhtml?fileId=4360331&version=2.0

For the training of the flow neural network, we employ EGNNs with 9 layers and 256 hidden features on QM9, and 4 layers and 256 hidden features on GEOM-DRUG, with a batch size of 64 and 16, respectively.

In the case of equivariant autoencoders, the decoder is parameterized in the same manner as the encoder, but the encoder is implemented with a 1-layer EGNN. This shallow encoder effectively constrains the encoding capacity and aids in regularizing the latent space (Xu et al., 2023).

All models utilize SiLU activations and are trained until convergence. Across all experiments, the Adam optimizer (Kingma & Ba, 2015) with a constant learning rate of $10^{-4}$ is chosen as our default training configuration. The training process for QM9 takes approximately 3000 epochs, while for GEOM-DRUG, it takes about 20 epochs.

With the flow model trained on QM9 or GEOM-DRUG, we then generate and purify the coupling to obtain a total of 100K molecular pairs, which form the estimated couplings.

**Hardware Configuration**

1. GPU: NVIDIA GeForce RTX 3090
2. CPU: Intel(R) Xeon(R) Platinum 8338C CPU
3. Memory: 512 GB
4. Time: Around 7 days for QM9 and 20 days for GEOM-DRUG.

## G    MORE EXPERIMENTAL RESULTS

We present the full results in Tables 5 and 6. In our detailed experimental results on QM9, we reproduced EDM, GeoLDM, and EquiFM on the QM9 dataset to obtain the actual generation time consumption with the same compute configuration. As a result, the proposed method achieves the fastest sampling speed, which is consistent with the measurement of sampling steps. We also witness a huge generation speed improvement by the proposed GOAT for GEOM-DRUG.

In addition to supplementing the actual time used for generation, we also added the metrics of molecule stability, and it is obvious that all methods achieve nearly 0% molecule stability in GEOM-DRUG. This is because metrics, atom and molecule stability, create errors during bond type prediction based on pair-wise atom types and distances. Therefore, we concentrate on metrics measured by RdKit.

Lastly, we produced the full results of GeoBFN using sampling steps from 50 to 1,000. It is worth noting that the novelty and significance continue to decrease on QM9 datasets as sampling steps increase, which aligns with our conjecture in the experiments. Besides, we also observed that its performance on GEOM-DRUG also decreased in terms of validity. Combined with its efficiency and quality, we believe that our method, GOAT, has competitive performance compared with GeoBFN.

We present the visualization of generated molecules on QM9 and GEOM-DRUG in Figures 5 and 6.

Table 5: Comparisons of generation quality (larger is better) in terms of Atom Stability, Molecule Stability, Validity, Uniqueness, Novelty, and Significance. And comparisons of generation efficiency regarding generation time and sampling steps for one molecule (less is better). The **best** results are highlighted in bold.

| # Metrics | QM9 | | | | | | | |
|---|---|---|---|---|---|---|---|---|
| | Efficiency | | Quality (%) | | | | | |
| | S-Time | Steps | Atom Sta | Mol Sta | Valid | Uniqueness | Novelty | Significance |
| Data | - | - | 99.0 | 95.2 | 97.7 | 100.0 | - | - |
| ENF | - | - | 85.0 | 4.9 | 40.2 | 98.0 | - | - |
| G-Schnet | - | - | 95.7 | 68.1 | 85.5 | 93.9 | - | - |
| GDM-aug | 1.50 | 1000 | 97.6 | 71.6 | 90.4 | **99.0** | 66.8 | 73.9 |
| EDM | 1.68 | 1000 | 98.7 | 82.0 | 91.9 | 98.7 | 65.7 | 64.8 |
| EDM-Bridge | - | 1000 | 98.8 | 84.6 | 92.0 | 98.6 | - | - |
| GeoLDM | 1.86 | 1000 | 98.9 | 89.4 | 93.8 | 98.8 | 58.1 | 53.9 |
| GeoBFN | - | 50 | 98.3 | 85.1 | 92.3 | 98.3 | 72.9 | 66.1 |
| | 0.16 | 100 | 98.6 | 87.2 | 93.0 | 98.4 | 70.3 | 64.4 |
| | - | 500 | 98.8 | 88.4 | 93.4 | 98.3 | 67.7 | 62.1 |
| | - | 1000 | **99.1** | **90.9** | **95.3** | 97.6 | 66.4 | 61.8 |
| EquiFM | 0.37 | 200 | 98.9 | 88.3 | 94.7 | 98.7 | 57.4 | 53.7 |
| GOAT | **0.12** | **90** | 98.4 | 84.1 | 90.0 | **99.0** | **78.6** | **72.3** |

Table 6: Comparisons of generation quality (larger is better) in terms of Atom Stability, Molecule Stability, Validity, Uniqueness, Novelty, and Significance. And comparisons of generation efficiency regarding generation time and sampling steps per molecule (less is better). The **best** results are highlighted in bold.

| # Metrics | GEOM-DRUG | | | | | |
|---|---|---|---|---|---|---|
| | Efficiency | | Quality (%) | | | |
| | S-Time | Steps | Atom Sta | Mol Sta | Valid | Uniqueness |
| Data | - | - | 86.5 | 0.0 | 99.9 | 100.0 |
| ENF | - | - | - | - | - | - |
| G-Schnet | - | - | - | - | - | - |
| GDM-aug | - | 1000 | 77.7 | - | 91.8 | - |
| EDM | 14.88 | 1000 | 81.3 | 0.0 | 92.6 | 99.9 |
| EDM-Bridge | - | 1000 | 82.4 | - | 92.8 | - |
| GeoBFN | - | 50 | 78.9 | - | 93.1 | - |
| | - | 100 | 81.4 | - | 93.5 | - |
| | - | 500 | 85.6 | - | 92.1 | - |
| | - | 1000 | **86.2** | - | 91.7 | - |
| GeoLDM | 12.84 | 1000 | 84.4 | 0.0 | **99.3** | 99.9 |
| EquiFM | - | 200 | 84.1 | - | 98.9 | - |
| GOAT | **0.94** | **90** | 84.8 | 0.0 | 96.2 | **99.9** |

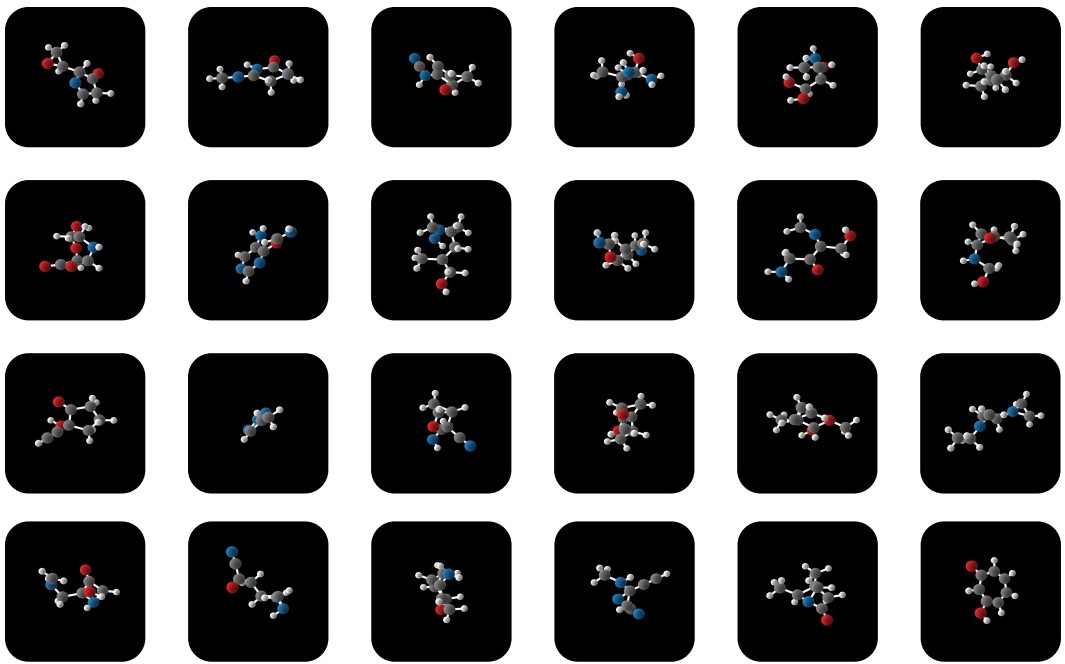

Figure 5: Molecules Generated by GOAT trained on QM9.

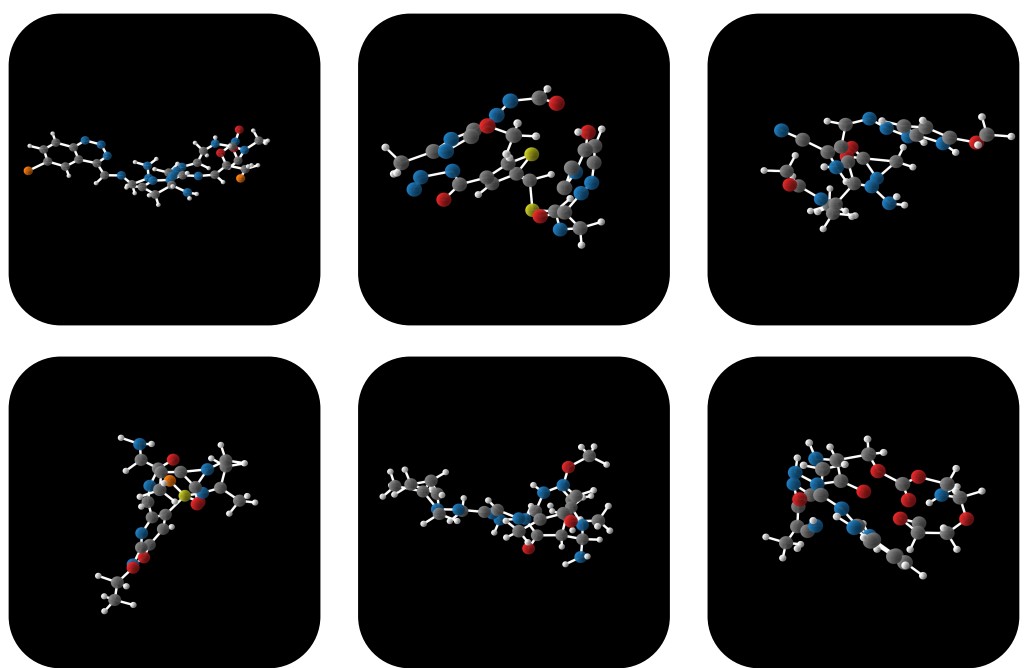

Figure 6: Molecules Generated by GOAT trained on GEOM-DRUG.

## H    DISTANCE BETWEEN NOISES AND GENERATED MOLECULES

We presented a comparison of the distance between generated molecules and the initial noise, including compared methods, the proposed GOAT, and its variants (w/o ODT and w/o OMT). We presented the experimental results in the Table:

Table 7: Distance Between Noises and Generated Molecules

| Metrics | QM9 | | GEOM-DRUG | |
|---|---|---|---|---|
| | Average distance | Average distance per atom | Average distance | Average distance per atom |
| EDM | 651.52 | 22.47 | 1834.22 | 10.13 |
| GeoLDM | 185.01 | 6.38 | 1046.67 | 5.78 |
| EquiFM | 530.86 | 18.31 | 1543.23 | 8.57 |
| GOAT-w/o ODT | 72.48 | 2.50 | 924.50 | 5.14 |
| GOAT-w/o OMT | 93.36 | 3.22 | 1190.88 | 6.62 |
| GOAT | **55.10** | **1.90** | **702.89** | **3.88** |

The experimental results on the distance between molecules and noises validate that the proposed method achieves the minimum transport distance from the noise and thereby also verifies the superiority of our method in generation speed.

## I    IMPACT STATEMENTS

This paper contributes to the advancement of generative Artificial Intelligence (AI) in scientific domains, including material science, chemistry, and biology. The insights gained will significantly enhance generative AI technologies, thereby streamlining the process of scientific knowledge discovery.

The application of machine learning to molecule generation expands the possibilities for molecule design beyond therapeutic purposes, potentially leading to the creation of illicit drugs or hazardous substances. This potential for misuse and unforeseen consequences underscores the need for stringent ethical guidelines, robust regulation, and responsible use of these technologies to safeguard individuals and society.

