# OpenReview forum: "Accelerating 3D Molecule Generation via Jointly Geometric Optimal Transport"
_ICLR.cc/2025/Conference — ICLR 2025 Poster_

### Official Review · Reviewer_N7NU · 2024-10-30

**Soundness:** 3
**Presentation:** 3
**Contribution:** 3
**Rating:** 6
**Confidence:** 3

**Summary:**

The paper presents a novel approach to molecule generation by introducing a variation of optimal transport for multimodal features within a general flow-matching objective. An equivariant neural network is utilized to transform these multimodal features into a latent space, where multimodal data optimal transport is applied. The results show that this method outperforms existing models like EquiFM and EDM, and it also demonstrates superior computational speed compared to other models.

**Strengths:**

1. **Significant Theoretical Development in Joint Optimal Transport:**

The theoretical advancement of joint optimal transport presented in the paper has the potential to greatly impact future flow-matching (FM)-based model development. This innovation could propel the entire field of molecule and conformer generation toward faster and more efficient methods.

2. **Emphasis on Performance Improvement and Comprehensive Comparisons:**

The paper places a strong focus on performance enhancement and provides thorough performance comparisons with models like EquiFM, EDM, and others. Since dataset conformations are obtained through computational methods, there is an upper bound on computational cost beyond which applying AI is not justified. The proposed model addresses this by improving performance while staying within acceptable computational limits.

**Weaknesses:**

1. **Comparison with Recent Edge-Modeling Methods and Potential Integration of Edge Features:**

Some recent models, such as JODO and EQGAT-Diff, have achieved better performance by explicitly modeling edges in molecular graphs. Including comparisons with these models would strengthen your paper. Moreover, adding edge features to your joint optimal transport framework seems feasible and could unlock the full potential of your method. Additionally, a more recent model called Semla Flow (a preprint published in June 2024) demonstrates superior accuracy and speed. Comparing your approach with these methods would greatly enhance the value of the proposed latent optimal transport (OT) technique.

2. **Concerns About Atom Stability Metric and 3D Evaluation:**

While the task is 3D molecule generation, there is no real evaluation of 3D coordinates for unconditional molecule generation in your paper. The term "atom stability" is mentioned without a clear definition. Given that you report an 86.5% figure for the GEOM Drugs dataset, I assume this refers to 3D atom stability. This metric is based on comparing bond lengths with tabulated values, allowing a tolerance of about 0.5 Å. The issue is that only 2.8% of GEOM Drugs molecules fully comply with these criteria, making the metric potentially misleading. I strongly encourage avoiding the propagation of this metric in new papers. The optimal distances between atoms are primarily defined by the potential energy landscape underlying the data—for GEOM Drugs, it's GFN2-xTB—and depending on atom configurations, deviations in bond lengths can exceed 10%.

3. **Difficulty in Claiming State-of-the-Art Performance:**

Related to the first point, it is difficult to claim that this method achieves state-of-the-art (SOTA) performance because it has not been compared with some other relevant methods mentioned above, including MiDi. Including such comparisons would strengthen the claim of achieving SOTA performance.

**Questions:**

1. Could you provide a more comprehensive evaluation of the generated 3D molecular structures?
I acknowledge that determining the most effective 3D metric is an open and important question. There are several possible solutions you might consider:
- The approach uses the Wasserstein distance between the distributions of bond lengths and bond angles. Some papers, like MiDi, have implemented this method. However, be cautious, as a single poorly predicted molecule can significantly skew the metric due to the way MiDi implemented it.
- SemlaFlow, for example, proposes using the optimization energy drop with the MMFF94 force field as a measure. Given that your ground truth data is based on GFN-xTB calculations, it might be more appropriate to assess the energy drop using GFN-xTB, as it aligns with the potential energy landscape of your dataset.
- Models like JODO and others have utilized MMD for bond angles, bond lengths, and torsion distributions, focusing only on the most frequent bonds and angles. While none of these metrics are perfect, they are generally more informative and insightful than relying solely on 3D atom stability.

The important part here is to ensure that when comparing models, you use exactly the same implementation of these metrics to obtain a reliable comparison. By incorporating these additional evaluations, you can provide a more thorough and insightful assessment of the geometric accuracy of your generated molecular structures. If I am adding one of these, I would rather do it on GEOM-Drugs dataset because it is much more realistic, for QM9 it is quite fast to compute GFN-xTB geometry optimization, and a lot of these molecules do not have considerable variability of 3D structure.

2. Could you perform comparisons with recent models like EQGAT-Diff, JODO, or SemlaFlow? I'm especially interested in a comparison with SemlaFlow because both papers place significant emphasis on improving the performance of molecule generation and utilize the flow matching objective. The main difference is that your approach performs flow matching in latent space, while SemlaFlow conducts diffusion explicitly on bonds, atoms, and coordinates.

It would be greatly appreciated if you could compare your method with SemlaFlow in terms of:
- Topological Metrics: Such as 2D molecule stability, validity, or introduced significance.
- 3D Metrics: Possibly using some of the metrics described in previous comments (e.g., bond angles, torsion distributions).
- Performance Metrics: Considering that both models generate molecules of similar size, metrics like the average time to generate a molecule would be insightful.

Additionally, could you discuss the advantages of your method over the version of flow matching used in SemlaFlow? Providing a brief discussion of the key algorithmic differences between your proposed method and SemlaFlow's flow matching approach—focusing on how these differences might impact performance or computational efficiency—would enhance the understanding of your contributions.

---

> ### Author Response · Authors · 2024-11-21
> **Response to Reviewer N7NU (1/2)**
>
> We appreciate the reviewer’s recognition of the strengths in our work. We are glad that the reviewer found our **significant theoretical development in joint optimal transport** and **performance improvement and comprehensive comparisons**. We address the reviewer's concerns below.
>
> # Response to the Weakness 2 and Question 1:
>
> > W2. Concerns About Atom Stability Metric and 3D Evaluation: ...
>
> > Q1. Could you provide a more comprehensive evaluation of the generated 3D molecular structures? ...
>
> We appreciate the reviewer's suggestions about the evaluation metrics. Following your suggestion, we supplemented the energy using GFN-xTB as an extra metric on the GEOM-DRUG dataset. We presented the supplemented results in the Table below:
>
> | GEOM-DRUG  | Quality  | ($\uparrow$) | Efficiency | ($\downarrow$) | Energy ($\downarrow$) |
> | ---------- | -------- | ------------- | ---------- | --------------- | ---------------------- |
> | Metrics    | Atom Sta | Valid         | Steps      | S-Time          | GFN2-xTB               |
> | Data       | 86.5     | 99.9          | \-         | \-              | \-2.034                |
> | GDM-aug    | 77.7     | 91.8          | 1000       | \-              |                        |
> | EDM        | 81.3     | 92.6          | 1000       | 14.88           | 8.0718                 |
> | EDM-Bridge | 82.4     | 92.8          | 1000       | \-              |                        |
> | GeoLDM     | 84.4     | __99.3__          | 1000       | 12.84           | __0.4954__                 |
> | GeoBFN     | 78.9     | 93.1          | 100        | 1.27            | 3.4396                 |
> | EquiFM     | 84.1     | 98.9          | 200        | 2.02            | 1.6964                 |
> | GOAT       | __84.8__     | 96.2          | __90__         | __0.94__            | 1.2745                 |
>
>
> The experimental results show that the energy obtained by the compared methods achieves competitive performance. __Among baselines focusing on fast molecule generation (GeoBFN and EquiFM), the proposed method GOAT achieves the lowest energy with faster sampling speed.__

---

> ### Author Response · Authors · 2024-11-21
> **Response to Reviewer N7NU (2/2)**
>
> # Response to Weakness 1, Weakness 3, and Question 2
>
> > W1. Comparison with Recent Edge-Modeling Methods and Potential Integration of Edge Features
>
> > W3. Difficulty in Claiming State-of-the-Art Performance ...
>
> > Q2. Could you perform comparisons with recent models like EQGAT-Diff, JODO ...
>
> We appreciate the reviewer's comment.
>
> First, We would like to highlight that the suggested algorithms are realized with different assumptions in terms of data modeling and experimental settings.
>
> 1) **Additional data features.** MiDi, EQGAT-Diff, JODO, and SemlaFlow all modeled bonds explicitly or implicitly, which required extra features of the molecular data.
>
> 2) **Filtered large molecules.** Particularly, these algorithms, including MiDi, EQGAT-Diff, and SemlaFlow, filtered out large-size molecules (more than 72 atoms) in the GEOM-DRUG dataset. Such assumption may limit their applicability in modeling edges for large-size molecules.
>
> 3) **Incomplete molecule modeling.** Many of them ignore the H atom in modeling the molecule, which also reduced the difficulty of the problem to some extent.
>
> It is unfair to claim that our methods are not SOTA as the data modeling assumptions and our experimental settings differ.
>
> Second, in response to Weakness 1, we implement our method with the integration of edge modeling and compare it to this type of method under the same experimental setting and datasets. We presented the results in the Table below:
>
> | QM9        |          |       | Quality    | ($\uparrow$) | | Efficiency   | ($\downarrow$) | Energy ($\downarrow$) |
> | ---------- | -------- | ----- | ---------- | ----------- | -- | ------------ | --------------- | ------ |
> | Metrics    | Atom Sta | Valid | Uniqueness | Novelty       | Significance | Steps           | S-Time | GFN2-xTB |
> | Data       | 99.0     | 97.7  | 100.0      | \-            | \-           | \-              | \-     | \-2.6049 |
> | MiDi       | 99.8     | 99.0  | 98.5       | 77.8          | 75.9         | 500             | 10.80  | \-2.4954 |
> | JODO       | __99.9__     | 99.0  | 97.0       | 74.2          | 71.2         | 1000            | \-     | \- |
> | EQGAT-Diff | 99.6     | 97.0  | __100.0__      | 73.4          | 71.2         | 500             | \-     | \- |
> | SemlaFlow  | __99.9__     | 99.4  | 97.5       | -          | -         | 100.0           | \-     | - |
> | GOAT-Bond  | 99.7     | __99.5__  | 99.5       | __78.6__          | __77.8__         | __90__              | __1.54__   | __-2.4130__ |
>
> Considering topological metrics, 3D metrics, and performance metrics, the proposed method with bond modeling also shows competitive performance compared to a series of methods that also model bonds, including MiDi, JODO, EQGAT-Diff, and SemlaFlow. Specifically, our method achieves the best validity, novelty, significance, sampling steps, and energy. We could not provide some results of JODO, EQGAT-Diff, and SemlaFlow, as some of them are not open-source, and the original paper did not report novelty and energy.
>
> Third, we would like to highlight the difference between our method and SemlaFlow.
>
> 1) **Equivariance.** We proposed a new equivariant optimal transport method for dealing with the multi-modal property of molecules. In contrast, SemlaFlow simply utilized existing equivariant optimal transport [1], both of which do not consider the multi-modality of molecules.
>
> 2) **Various sizes of molecules.** To deal with various sizes of molecules, we proposed the flow refinement and purification mechanism, whereas SemlaFlow utilized several techniques, like layer normalization, to circumvent this challenge.
> 3) **Conditional generation.** SemlaFlow did not provide experiments on conditional generation. In contrast, our method achieves a better trade-off between accuracy and efficiency in conditional molecule generation.
>
> [1] Song, Yuxuan, et al. "Equivariant flow matching with hybrid probability transport for 3d molecule generation." Advances in Neural Information Processing Systems 36 (2023).

---

> > ### Comment · Reviewer_N7NU · 2024-11-21
> > **Preliminary response to the provided detailes**
> >
> > Thank you very much for the detailed answer!
> >
> > I am fully satisfied with your highlighting the differences between your approach and Semla-Flow.
> >
> > I appreciate you adding the GFN2-xTB geometry optimization energy drop as a metric. Can you please add units for the metric (I think the best is kcal/mol)? Also, for some reason, you did not get results close to zero for the dataset itself, though it should be around zero because of the procedure. The optimization energy drop should be positive (basically, you minimize the energy function with respect to coordinates and compute the difference between the initial value and the minimized value). I know the xTB interface can be confusing, so I attach a code snippet of how to parse the value from the xTB output:
> > ```
> > def parse_xtb_output(xtb_output):
> >     """Parse xTB output to get total energy gain and total RMSD."""
> >     total_energy_gain = None
> >     total_rmsd = None
> >
> >     lines = xtb_output.splitlines()
> >     for line in lines:
> >         if "total energy gain" in line:
> >             total_energy_gain = float(line.split()[6])  # in kcal/mol
> >         elif "total RMSD" in line:
> >             total_rmsd = float(line.split()[5])  # in Angstroms
> >     return total_energy_gain, total_rmsd
> > ```
> >
> > There are a few other inaccuracies in your response when commenting on other models. All the mentioned models work with all-atom molecules, including hydrogens. Also, as far as I know, only Semla-Flow filtered the dataset by molecule size.

---

> > > ### Author Response · Authors · 2024-11-26
> > > **Response to the Reviewer N7NU**
> > >
> > > We appreciate your prompt and detailed feedback.
> > >
> > > # Response to the energy metric
> > > The unit for the metric is Hartree, and the reported energy is indeed calculated from single-point results. We appreciate the code snippet you provided and have updated the results to include the energy drop and RMSD after optimization.
> > >
> > > ## Updated experimental results on the energy drop using GFN-xTB on the GEOM-DRUG
> > > | GEOM-DRUG  | Quality  | ($\\uparrow$) | Efficiency | ($\\downarrow$) | GFN2-xTB                 | ($\\downarrow$)        |      |
> > > | ---------- | -------- | ------------- | ---------- | --------------- | ------------------------ | ------------------------- | --- |
> > > | Metrics    | Atom Sta | Valid         | Steps      | S-Time          | Energy Single Point (Eh) | Total Energy Drop (kcal/mol) | Total RMSD (Å) |
> > > | Data       | 86.5     | 99.9          | \-         | \-              | \-2.034                  | \-                           | \- |
> > > | GDM-aug    | 77.7     | 91.8          | 1000       | \-              | \-                       | \-                           | \- |
> > > | EDM        | 81.3     | 92.6          | 1000       | 14.88           | 8.0718                   | 456.35                       | 1.89 |
> > > | EDM-Bridge | 82.4     | 92.8          | 1000       | \-              | \-                       | \-                           | \- |
> > > | GeoLDM     | 84.4     | __99.3__          | 1000       | 12.84           | __0.4954__                   | __177.03__                       | __1.41__ |
> > > | GeoBFN     | 78.9     | 93.1          | 100        | 1.27            | 3.4396                   | 360.59                       | 1.60 |
> > > | EquiFM     | 84.1     | 98.9          | 200        | 2.02            | 1.6964                   | 227.84                       | 1.55 |
> > > | GOAT       | __84.8__     | 96.2          | __90__         | __0.94__            | 1.2745                   | 189.61                       | 1.45 |
> > >
> > > The experimental results indicate that our algorithm exhibits competitive performance regarding total energy drop and total RMSD. __Among the baselines prioritizing rapid molecule generation, specifically GeoBFN and EquiFM, our proposed method, GOAT, achieves the lowest energy drop while also demonstrating a faster sampling speed.__
> > >
> > > ## Updated experimental results on the integration of edge modeling
> > > | QM9        |          |       | Quality    | ($\\uparrow$) | | Efficiency   | ($\\downarrow$) | GFN2-xTB | ($\\downarrow$)        |  |
> > > | ---------- | -------- | ----- | ---------- | ------------ | - | ------------ | --------------- | -------- | --------------------- | --- |
> > > | Metrics    | Atom Sta | Valid | Uniqueness | Novelty       | Significance | Steps           | S-Time   | Energy Single Point (Eh) | Total Energy Drop (kcal/mol) | Total RMSD (Å) |
> > > | Data       | 99.0     | 97.7  | 100.0      | \-            | \-           | \-              | \-       | \-2.6049                 | \- | \- |
> > > | MiDi       | 99.8     | 99.0  | 98.5       | 77.8          | 75.9         | 500             | 10.80    | \-2.4954                 | 5.7534 | 0.2691 |
> > > | JODO       | __99.9__     | 99.0  | 97.0       | 74.2          | 71.2         | 1000            | \-       | \-                       | \- | \- |
> > > | EQGAT-Diff | 99.6     | 97.0  | __100.0__      | 73.4          | 71.2         | 500             | \-       | \-                       | \- | \- |
> > > | SemlaFlow  | __99.9__     | 99.4  | 97.5       | 68.5          | 66.4         | 100.0           | \-       | \-                       | \- | \- |
> > > | GOAT-Bond  | 99.7     | __99.5__  | 99.5       | __78.6__          | __77.8__         | __90__              | __1.54__     | __\-2.4130__                 | __3.9558__ | __0.2471__ |
> > >
> > > The updated tables demonstrate that our framework results in a reduced energy drop and lower RMSD compared to MiDi. These findings suggest that the molecules produced by our methods are closer to the optimized geometry obtained using GFN2-xTB.
> > >
> > > We acknowledge the ambiguity in our previous response and thank you for pointing out these issues.

---

> > > > ### Comment · Reviewer_N7NU · 2024-11-28
> > > > **Rebuttle response**
> > > >
> > > > Thank you for your comprehensive answer!
> > > >
> > > > The authors addressed all of the comments. I especially appreciate that they added the GOAT-Bond and energy benchmarks. However, I would like to ask them to verify if the energy drop is correct for the GEOM Drugs dataset, as the values are on the order of hundreds of kcal/mol, which is quite high for molecules (the typical bond breaking energy is around 100 kcal/mol for comparison). I have adjusted my score to positive based on their response.

---

> > > > > ### Author Response · Authors · 2024-12-01
> > > > > **Response to Reviewer N7NU**
> > > > >
> > > > > We thank the reviewer for the constructive comments and sincerely appreciate the reviewer's recognition of our response.
> > > > >
> > > > > # Response to the energy drop
> > > > >
> > > > > > I would like to ask them to verify if the energy drop is correct for the GEOM Drugs dataset, as the values are on the order of hundreds of kcal/mol, which is quite high for molecules (the typical bond breaking energy is around 100 kcal/mol for comparison).
> > > > >
> > > > > We acknowledge the reviewer's concern regarding the high energy drop values for the GEOM-DRUG dataset. We have thoroughly verified our results.
> > > > >
> > > > > Specifically, we utilized `xtb` version 6.6.1 (8d0f1dd). The generated molecules were saved in `molecule.xyz` files and the command `xtb molecule.xyz --opt` was executed to obtain the `xtb` output. Subsequently, we employed the suggested code provided by the reviewer to extract the `total energy gain` and `total RMSD`.
> > > > >
> > > > > For the reviewer's reference, we have randomly selected some `.xyz` files and `xtb` outputs and uploaded them via an anonymous [link](https://drive.google.com/file/d/1Jqt1yGVLqSg4vP4LlCngKwkDbHbmZIVq/view?usp=share_link).

---

### Official Review · Reviewer_cmjL · 2024-11-01

**Soundness:** 3
**Presentation:** 3
**Contribution:** 3
**Rating:** 6
**Confidence:** 4

**Summary:**

This paper introduces a novel framework for accelerating 3D molecule generation, named GOAT, which leverages the principles of flow-matching optimal transport to efficiently generate molecules with improved quality and speed.

**Strengths:**

1. I think that using optimal transport to optimize the training process of diffusion models is a very clever and reasonable design, especially when applied to scenarios like molecular generation, where the generation efficiency of diffusion models can be a concern.
2. I'm glad that the paper points out the optimal transport optimization might bring extra computational cost, especially those involving rotation and permutation, can be computationally intensive. However , it's inevitable, but it's a great trial.
3. The framework achieves the satisfiable downstream generation quality regarding validity, uniqueness, and novelty, which are crucial metrics in molecule generation.

**Weaknesses:**

1. While the paper includes some ablation studies, a more comprehensive set of experiments could provide further insights into the contribution of each component of the framework.
2. The training process itself may be time-consuming, which could be a drawback for some applications, though as mentioned in strengths, it's inevitable.

**Questions:**

1. It would be interesting if the paper could provide a comparison of the distance between generated molecules and the initial state, with/without the use of optimal transport.
2. The use of optimal transport may raise concerns about the diversity of the generated molecules. This could be addressed by generating N molecules and comparing the coverage of the reference conformation set within a specified distance threshold (you can refer to the design of this criterion in papers like GeoDiff[1]).
[1] Xu, Minkai, et al. "Geodiff: A geometric diffusion model for molecular conformation generation"

---

> ### Author Response · Authors · 2024-11-21
> **Response to Reviewer cmjL**
>
> We are grateful that the reviewer acknowledges the strengths of our work. We are glad that the reviewer found that **our design is very clever and reasonable**, **discussion on the computational cost is comprehensive**, and **our approach achieves a satisfiable performance**. We address the reviewer's concerns below.
>
> # Response to Weakness 1
> > While the paper includes some ablation studies ...
>
>
> In response to your concern, we added experiments evaluating the significance of the equivariant autoencoder (w/o EAE). The new results are added in row (__w/o EAE__) below:
>
> | Metrics (QM9) | | | Quality | ($\uparrow$) | | Efficiency | ($\downarrow$) | |
> | ------------- | -------- | ----- | --------------------- | ------- | ------------ | -------------------------- | ---- | ---- |
> | Components| Atom Sta | Valid | Uniqueness| Novelty | Significance | Steps| Time | Cost |
> | __w/o EAE__ | _98.8_ | _92.8_| 92.2| 58.4| 53.9 | 280| 0.63 | 30.25 |
> | w/o ODT | 98.1 | 89.9| 98.8| _70.4_| _62.5_ | _100_| _0.14_ | _4.13_ |
> | w/o OMT | 96.5 | 85.0| _98.9_| 69.1| 58.1 | 170| 0.31 | 5.32 |
> | GOAT| __99.2__ | __92.9__| __99.0__| __78.6__| __72.3__ | __90.0__ | __0.12__ | __3.14__ |
>
> The findings indicate that without the equivariant autoencoder (w/o EAE), the transport cost is significantly higher, and the generation time nearly quadruples compared to the GOAT framework. This underscores the importance of jointly optimizing transport for atom coordinates and features.
>
> # Response to Weakness 2
> > The training process itself may be time-consuming...
>
> We agree with the reviewer’s comment, which we have acknowledged in our manuscript. We will investigate methods to enhance the training efficiency of our framework.
>
> # Response to Question 1
> > It would be interesting if the paper could provide a comparison...
>
> We thank the reviewer for the insightful suggestions. We presented a comparison of the distance between generated molecules and the initial noise, including compared methods, the proposed GOAT, and its variants (w/o ODT and w/o OMT). We presented the experimental results in the Table:
>
> |              | QM9              |                           | GEOM-DRUG        | |
> | ------------ | ---------------- | ------------------------- | --------------|-- |
> | Metrics      | Average distance | Average distance per atom | Average distance | Average distance per atom |
> | EDM          | 651.52           | 22.47                     | 1834.22          | 10.13 |
> | GeoLDM       | 185.01           | 6.38                      | 1046.67          | 5.78 |
> | EquiFM       | 530.86           | 18.31                     | 1543.23          | 8.57 |
> | GOAT-w/o ODT | 72.48            | 2.50                      | 924.50           | 5.14 |
> | GOAT-w/o OMT | 93.36            | 3.22                      | 1190.88          | 6.62 |
> | GOAT         | __55.10__            | __1.90__                      | __702.89__           | __3.88__ |
>
> The experimental results on the distance between molecules and noises validate that the proposed method achieves the minimum transport distance from the noise and thereby also verifies the superiority of our method in generation speed.
>
> # Response to Question 2
> > The use of optimal transport may raise concerns about ...
>
> We appreciate the reviewer’s valuable suggestions and agree that diversity is a crucial evaluation metric. However, we note that the COV-R and MAT-R metrics focus specifically on conformation generation tasks, where the accuracy of generated conformations is assessed against a set of reference conformers. In our molecular generation task, there are no predefined target conformations. Utilizing a training set would result in evaluating the generation of existing molecules, which contradicts the objective of de novo drug design. Consequently, preparing a reference set is challenging.
>
> Nonetheless, our adopted metric of uniqueness addresses this concern. Uniqueness measures the fraction of unique molecules among all valid generated samples. In our evaluation with 10,000 samples, the proposed method achieved an impressive 99.0\% uniqueness, indicating that approximately 9,900 distinct molecules were generated, demonstrating our method's capability to produce diverse molecular structures.
>
> We recognize that our explanation may not completely cover your comment. If you could share more details, we would be happy to provide a more focused response.

---

### Official Review · Reviewer_AHDV · 2024-11-01

**Soundness:** 3
**Presentation:** 2
**Contribution:** 3
**Rating:** 6
**Confidence:** 3

**Summary:**

This paper proposes a 3D molecule generation framework which formulates a geometric transport formula for
measuring the cost of mapping multi-modal features (e.g., continuous atom coordinates and categorical atom types) between a base distribution and a target data distribution. They further propose a flow refinement and purification mechanism for optimal coupling identification, which filters out the subpar molecules to ensure the ultimate generation quality.

**Strengths:**

1. The problem setup is well-motivated. The OT path and equivariance properties are indeed needed in diffusion models/flow matching methods for fast sampling process.

2. The empirical performance is strong in terms of various metrics. The proposed methods is good in both generation quality and inference steps.

**Weaknesses:**

1. In Eq. (5), the authors seem to sample permutation, rotation, and translation matrix. Does this mean that the model is not strictly equivariant but approximately equivariant? In other words, the equivariance is learned from data augmentation but not learned by construction.

2. I am wondering maybe it is possible to distill the trained model, like rectified flow, to have even faster inference (1 to 5 steps). I am especially curious to see the generation quality of distillation + purification.

**Questions:**

Please refer to weaknesses.

I honestly do not know much about the literature on molecule generation so my confidence would not be high. I am happy to check other reviewers' opinions.

---

> ### Author Response · Authors · 2024-11-21
> **Response to Reviewer AHDV**
>
> We appreciate that the reviewer acknowledges the strengths of our work. We are glad that the reviewer found that our **motivation is well-defined and strong** and **the performance is strong as well**. We address the reviewer's concerns below.
>
>
> # Response to Weakness 1:
> > In Eq. (5), the authors seem to sample permutation...
>
> Thank you for your insightful question. We somewhat agree with the reviewer that the optimized permutation, rotation, and translation matrices derived from Eq. (5) serve as a means of data augmentation [1].
> As discussed in [1], data augmentation can somehow eliminate the equivariance requirement in model design. In addition, our model achieves a strict equivariance with an equivariant graph neural network.
>
> Specifically, we can first establish that the latent data $z_1$ is equivariant with respect to the data point $\mathbf{g}_1$ as implemented in the equivariant autoencoder.
>
> Following the approach in [2], we then show that the training loss in Eq. (8) is invariant to translations and rotations of the noise $z_{0}$ and equivariant with respect to both the sampled noise $z_{0}$ and the latent data point $z_{1}$. Therefore, the optimized permutation, rotation, and translation do not compromise the model's equivariance.
>
> [1] Brehmer, Johann, et al. "Does equivariance matter at scale?." arXiv preprint arXiv:2410.23179 (2024).
> [2] Song, Yuxuan, et al. "Equivariant flow matching with hybrid probability transport for 3d molecule generation." Advances in Neural Information Processing Systems 36 (2023).
>
> # Response to Weakness 2:
> > I am wondering maybe it is possible to distill the trained model...
>
> We appreciate the reviewer’s insightful comments. We conducted experiments to evaluate the performance of the distilled model using the distillation method introduced in Rectified Flow, and the results are presented below.
>
> | QM9            |          |       | Quality    | ($\uparrow$) | | Efficiency   | ($\downarrow$) |
> | -------------- | -------- | ----- | ---------- | -----------|-- | ------------ | --------------- |
> | Metrics        | Atom Sta | Valid | Uniqueness | Novelty       | Significance | Steps           | S-Time |
> | Data           | 99.0     | 97.7  | 100.0      | \-            | \-           | \-              | \- |
> | ENF            | 85.0     | 40.2  | 98.0       | \-            | \-           | \-              | \- |
> | G-Schnet       | 95.7     | 85.5  | 93.9       | \-            | \-           | \-              | \- |
> | GDM-aug        | 97.6     | 90.4  | __99.0__       | 74.6          | 66.8         | 1000            | 1.50 |
> | EDM            | 98.7     | 91.9  | 98.7       | 65.7          | 59.6         | 1000            | 1.68 |
> | EDM-Bridge     | 98.8     | 92.0  | 98.6       | \-            | \-           | 1000            | \- |
> | GeoLDM         | 98.9     | 93.8  | 98.8       | 58.1          | 53.9         | 1000            | 1.86 |
> | GeoBFN         | 98.6     | 93.0  | 98.4       | 70.3          | 64.4         | 100             | 0.16 |
> | EquiFM         | 98.9     | __94.7__  | 98.7       | 57.4          | 53.7         | 200             | 0.37 |
> | GOAT           | __99.2__     | 92.9  | __99.0__       | 78.6          | __72.3__         | 90              | 0.12 |
> | GOAT-distilled | 60.6    | 12.2  | 11.0      | __99.3__        | 1.34         | __1__               | __0.07__ |
>
> The experimental results indicate that distillation improves generation speed; the distilled model achieves the best S-Time. Although the validity is lower, the distilled model produces valid, unique, and novel molecules at the fastest speed. However, the stability and validity of the atoms remain unsatisfactory. We conjecture that the distillation on molecular generators requires additional designs, and we plan to investigate the distillation method in molecular generation further in our future research.

---

> > ### Comment · Reviewer_AHDV · 2024-11-25
> > **Thanks for the response**
> >
> > Thanks for the response. My concern has been solved.

---

### Official Review · Reviewer_DEWZ · 2024-11-02

**Soundness:** 2
**Presentation:** 3
**Contribution:** 2
**Rating:** 6
**Confidence:** 3

**Summary:**

Because of the known multi-modal (Cartesian coordinates and other features) distribution issue in molecule generation and distribution coupling issue, the paper proposed an optimal molecule transport (OMT) algorithm based on optimal transport (OT) in Flow Matching (FM) on latent codes encoded by equivariance autoencoder.

**Strengths:**

1. **Clear problem statement**: optimal transport of multi-modal probability and distribution coupling in the specific case of molecule generation are well-known problems and to be addressed. The writing provides enough information for general readers to comprehend.
The experiments seem abundant to make the effectiveness of the method stand.
2. Table 4 is crucial for the core statement of this paper, which is that Optimal Molecule Transport (OMT) is important for faster and better molecule generation based on optimal transport (OT) of Flow Matching (FM). From the numbers there, it seems justified.

**Weaknesses:**

1. I do not see why after equivariance autoencoder mapping to the latent space, we thus have a unified optimal permutation. In equation (5), we still have the same \pi for both coordinates and features right? The reasoning for this part is missing.
2. I am not sure the distribution coupling, $\Gamma$ is a correct terminology used in this paper. The distribution coupling should be referring to pairing points between two distributions, rather than pairing within one pair of data sample by permutation, rotation and translation.
3. Maybe limited novelty: In EquiFM[1], we know the idea of equivariant optimal transport (EOT) is proposed, which has a large overlapping with the core contributions of this paper. Can this paper’s main contributions be concluded as  EOT + Equivariant Autoencoder? If so, the novelty of paper may be limited.
4. Is Geometric Probability Distribution a formal terminology in molecule generation community? I do not see it is widely adopted in other related work and it can be misunderstood with geometric distribution.
5. The abbreviation EAE in line 251 pops out without specification (I am assuming it is referring to Equivaraint Autoencoder?)

[1] Song, Yuxuan, et al. "Equivariant flow matching with hybrid probability transport for 3d molecule generation." Advances in Neural Information

**Questions:**

Based on the listed weakness points above, I want to ask following questions:

1. Why using equivariance autoencoder to produce latent codes can yield in unified permutation?
2. Can I assume your paper is based on the already on-the-shelf work which proposed EOT and EAE? If so, what is your major novel improvement scientifically based on that? If not, please clarify why.
3. Please clarify the terminology issues on point 2 and point 4 mentioned above in the weakness part.

I will adjust my rating to the paper based on the answers to the above questions during rebuttal and discussion phase.

---

> ### Author Response · Authors · 2024-11-21
> **Response to Reviewer DEWZ**
>
> We are glad that the reviewer recognizes the strengths of our work. We appreciate the reviewer found that **our problem statement is clear**, **the entire writing is comprehensive**, and **our experiments are abundant to illustrate the effectiveness**. We address the reviewer's concerns as follows.
>
> # Response to Weakness 1
> > W1. In equation (5), we still have the same $\pi$ for both coordinates and features right? The reasoning for this part is missing.
>
> Thanks for the astute observation. Coordinates and features together depict the atomic-level geometry of a molecule. Therefore, a unified optimal $\pi$ for both coordinates and features is needed for obtaining an optimal transport, leading to fast and high-quality molecule generation.
>
> # Response to Weakness 1 and Question 1
> > W1. I do not see why after equivariance autoencoder mapping to the latent space, we thus have a unified optimal permutation.
>
> > Q1. Why using equivariance autoencoder to produce latent codes can yield in unified permutation?
>
> Thanks for the insightful question. We would like to clarify that the equivariance autoencoder is designed to deal with the multi-modal properties of molecules, mapping coordinates and atom features into a unified, continuous latent space. Simply mapping with the equivariance autoencoder indeed could not directly yield a unified optimal permutation. It is critical to achieve an optimal permutation to minimize the transport cost; we realize such an objective using the Hungarian algorithms.
>
>
> # Response to Weakness 2, 4, and Question 3:
> > W2. I am not sure the distribution coupling, $\Gamma$ is ...
>
> > W4. Is Geometric Probability Distribution a formal terminology ...
>
> > Q3. Please clarify the terminology issues ...
>
> Thanks for pointing out the terminology issues. We address the reviewer's comments one by one below.
>
> 1) In response to W2 \& Q3: we specify the notation $\Gamma$ for the pairing points between two molecule distributions. Specifically, we use $(\mathbf{g}_0,\mathbf{g}_1)$ to denote the pairing points, where $\mathbf{g}_0$ is sampled from the noise distribution and $\mathbf{g}_1$ is sampled from the data distribution. As in Eq. (5), we consider permutation, rotation, and translation for geometric optimal transport cost; the purpose is to preserve geometric symmetries while minimizing the transport cost.
>
> 2) In response to W4 \& Q3: The purpose of using geometric probability distribution is to reflect the geometric property of molecules. We are unaware of introducing such confusion to the readers. For clarity, we have added an explanation of this terminology in the manuscript (a footnote on page 3).
>
> We understand that our explanation might not fully address your comment. You could provide further details or specify the aspects that are unclear. We would be glad to further offer a more targeted explanation.
>
> # Response to Weakness 3 and Question 2:
> > Maybe limited novelty: In EquiFM[1], …
>
> > Can I assume your paper is based on …
>
> We thank the reviewer for giving us the chance to re-clarify our novelty. **Simply combining EOT and Equivariant Autoencoder does not work. Therefore, our main contributions could not be concluded as EOT + Equivariant Autoencoder.** Firstly, EOT (or EquiFM [1]) indeed attempts to accelerate the generation process with the notion of flow matching. However, it didn't consider the multi-modal properties of molecules, leading to sub-optimal transport. Different from EquiFM [1], we consider both modalities by mapping the continuous atom coordinates and categorical atom types with the Equivariant Autoencoder. Secondly, though with the mapping, searching for optimal coupling is still challenging due to the various sizes of 3D molecules. To address the challenges, we propose flow refinement and purification to approximate optimal transport and accelerate molecule generation.
>
> [1] Song, Yuxuan, et al. "Equivariant flow matching with hybrid probability transport for 3d molecule generation." Advances in Neural Information Processing Systems 36 (2023).
>
> # Response to Weakness 5:
> > The abbreviation EAE in line 251…
>
> Yes. EAE represents Equivaraint Autoencoder, which we have updated on page 5 of the manuscript.

---

> > ### Comment · Reviewer_DEWZ · 2024-11-26
> >
> > Hi, thanks for your response. Based on your response, I have few further questions:
> >
> > 1. It seems like the unified $\pi$ comes from the adoption of EAE. Is this true?
> > 2. About the novelty issue of your response to **W3** and **Q2**, the first half of your response does seem like exactly EOT + EAE. And in the second half of your reponse, you mention propose something new called flow refinement and purifaction. I searched for the term `flow refinement`, I only found it in the first half of the manuscript. When I searched for `flow refine`, there appeared to be one more on page 6, which indicates it was a result induced by the `the estimated optimal coupling`. I have to complain that the terminologies in this submission have been made unncessary complicated. Moreover, the spirit of optimal coupling here seems to be aligned with the one in EOT as well. Do I understand this correctly?

---

> > > ### Author Response · Authors · 2024-12-01
> > > **Response to Reviewer DEWZ**
> > >
> > > We appreciate your prompt response and insightful comments.
> > >
> > > # Response to Question 1.
> > > > It seems like the unified $\pi$ comes from the adoption of EAE. Is this true?
> > >
> > > We would like to clarify that unified $\pi$ did not directly come from the adoption of EAE, but EAE is the basis for obtaining the unified $\pi$. To obtain the unified $\pi$, our framework first mapped atom coordinates and features into the unified space by EAE. Second, after the mapping, the unified $\pi$ is achieved based on the Hungarian algorithm.
> > >
> > > # Response to Question 2.
> > >
> > > ## Response to the terminology
> > >
> > > > About the novelty issue of your response to W3 and Q2, the first half of your response does seem like exactly EOT + EAE. And in the second half of your reponse, you mention propose something new called flow refinement and purifaction. I searched for the term flow refinement, I only found it in the first half of the manuscript. When I searched for flow refine, there appeared to be one more on page 6, which indicates it was a result induced by the the estimated optimal coupling. I have to complain that the terminologies in this submission have been made unncessary complicated.
> > >
> > > We apologize for the confusion of terminologies. The term `flow refinement and purification` refers to our proposed method for estimating optimal coupling and filtering out invalid molecules to purify the coupling. For clarity, we have revised `flow refinement` to `optimal coupling estimation`. Please refer to our revised manuscript.
> > >
> > > ## Response to the spirit of optimal coupling
> > >
> > > > Moreover, the spirit of optimal coupling here seems to be aligned with the one in EOT as well. Do I understand this correctly?
> > >
> > > We agree with the reviewer that the spirit of optimal coupling aligns with the one in EOT, as both of them are proposed to address the optimal transport problem. However, we would like to identify the differences between optimal coupling and EOT:
> > >
> > > 1. EOT (or EquiFM [1]) focuses on searching for the optimal transport path for the coordinates between two molecules that are coupled randomly.
> > > 2. Optimal coupling in our paper concentrates on searching for optimal coupling between noise distribution and molecule data distribution.
> > >
> > > [1] Song, Yuxuan, et al. "Equivariant flow matching with hybrid probability transport for 3d molecule generation." Advances in Neural Information Processing Systems 36 (2023).

---

> ### Comment · Reviewer_DEWZ · 2024-12-02
>
> Thanks to the authors for their clarifications. I maintain my rating of leaning toward acceptance.

---

### Author Response · Authors · 2024-11-21
**General Response**

We sincerely appreciate the time and efforts of the reviewers in providing their valuable feedback.

The key points of our rebuttal can be summarized as follows:

1. Provide extensive discussion and clarifications of the following aspects

> 1.1. The novelty of our work ([link](https://openreview.net/forum?id=VGURexnlUL&noteId=VnB6Do1I7Y))

> 1.2 The use of some terminologies ([link](https://openreview.net/forum?id=VGURexnlUL&noteId=VnB6Do1I7Y))

> 1.3 The equivariance justification of our model ([link](https://openreview.net/forum?id=VGURexnlUL&noteId=vMQU1f6J0L))

> 1.4 The diversity of our generated molecules ([link](https://openreview.net/forum?id=VGURexnlUL&noteId=RLL8sryCdV))

2. Conduct additional experiments on the following aspects:

> 2.1 The performance of our distilled model ([link](https://openreview.net/forum?id=VGURexnlUL&noteId=vMQU1f6J0L))

> 2.2 The effectiveness of equivariance autoencoder ([link](https://openreview.net/forum?id=VGURexnlUL&noteId=RLL8sryCdV))

> 2.3 The distance between generated molecules and the initial noise samples ([link](https://openreview.net/forum?id=VGURexnlUL&noteId=RLL8sryCdV))

> 2.4 The extra evaluation of energy ([link](https://openreview.net/forum?id=VGURexnlUL&noteId=SfIEQIq56g))

> 2.5 The comparison with bond-modeled methods ([link](https://openreview.net/forum?id=VGURexnlUL&noteId=pSd20mUUjY))

---

### Meta-Review · Area_Chair_VnYk · 2024-12-19

**Metareview:**

This paper introduces GOAT, a novel 3D molecule generation framework leveraging joint geometric optimal transport and equivariant autoencoders, which reviewers found innovative and effective in addressing multi-modal molecule properties while achieving impressive generation quality and speed. Strengths highlighted include clear motivation, significant theoretical contributions to flow matching methods, and competitive empirical performance, with notable efficiency gains over baseline models. However, reviewers raised concerns about limited comparisons to recent edge-modeling methods, potential ambiguities in terminology, and the adequacy of certain evaluation metrics, suggesting room for improvement in comprehensive benchmarking and clarity of presentation.

**Additional Comments On Reviewer Discussion:**

During the rebuttal, reviewers raised concerns about the novelty of GOAT compared to existing methods like EquiFM, the adequacy of evaluation metrics for 3D molecule generation, and the lack of comparisons with recent edge-modeling approaches such as JODO and SemlaFlow. The authors clarified their novel contributions, enhanced evaluation by incorporating energy drop and RMSD metrics using GFN-xTB, and added experiments with edge modeling to address these comparisons, demonstrating GOAT’s competitive performance. Additionally, they addressed ambiguities in terminology and justified their methodological choices, including the integration of equivariance and flow refinement, which satisfied most reviewer concerns and led to an overall positive consensus.

---

### Decision · Program_Chairs · 2025-01-22

Accept (Poster)